



# Identifying atmospheric processes favouring the formation of bubble free layers in Law Dome ice core, East Antarctica

Lingwei Zhang[1,2], Tessa R. Vance[2], Alexander D. Fraser[2], Lenneke M. Jong[3,2], Sarah S. Thompson[2], Alison Criscitiello[4], and Nerilie J. Abram[5]

[1]Institute for Marine & Antarctic Studies, University of Tasmania, Battery Point 7004, Tasmania Australia
[2]Australian Antarctic Program Partnership, Institute for Marine & Antarctic Studies, University of Tasmania, Battery Point 7004, Tasmania Australia
[3]Australian Antarctic Division, Channel Highway, Kingston 7050, Australia
[4]Department of Earth and Atmospheric Sciences, University of Alberta, Edmonton T6G 2R3, Canada
[5]Research School of Earth Sciences and ARC Centre of Excellence for Climate Extremes, Australian National University, Canberra ACT 2601, Australia

**Correspondence:** Lingwei Zhang (Lingwei.zhang@utas.edu.au)

**Abstract.**

   Physical features preserved in ice cores may provide unique records about past atmospheric variability. Linking the formation and preservation of these features and the atmospheric processes causing them is key to their interpretation as paleoclimate proxies. We imaged ice cores from Law Dome, East Antarctica using an Intermediate Layer Ice Core Scanner (ILCS) which

shows that thin bubble-free layers (BFLs) occur multiple times per year at this site. The origin of these features is unknown. We used a previously developed age-depth scale in conjunction with regional accumulation estimated from atmospheric reanalysis data (ERA5) to estimate the year and month that the BFLs occurred, and then performed seasonal and annual analysis to reduce the overall dating errors. We then investigated measurements of snow surface height from a co-located automatic weather station to determine snow surface features co-occurring with BFLs, as well as their estimated occurrence date. We

also used ERA5 to investigate potentially relevant local/regional atmospheric processes (temperature inversions, wind scour, accumulation hiatuses and extreme precipitation) associated with BFL occurrence. Finally, we used a synoptic typing dataset of the southern Indian and southwest Pacific Oceans to investigate the relationship between large scale atmospheric patterns and BFL occurrence. Our results show that BFLs occur (1) primarily in autumn and winter, (2) in conjunction with accumulation hiatuses >4 days, and (3) during synoptic patterns characterised by meridional atmospheric flow related to the episodic blocking

and channeling of maritime moisture to the ice core site. Thus, BFLs may act as a seasonal marker (autumn/winter), and may indicate episodic changes in accumulation (such as hiatuses) associated with large-scale circulation. This study provides a pathway to the development of a new proxy for past climate in the Law Dome ice cores; specifically past snowfall conditions relating to synoptic variability over the southern Indian Ocean.



## 1 Introduction

Ice cores provide one of the most powerful tools available to determine how the Earth's climate has varied in the past. Records of climate-active gases preserved in ice cores have provided the $CO_2$ concentration over the past 800,000 years (Etheridge et al., 1996; Rubino et al., 2013). Trace chemical impurities and water isotopic ratios in polar ice vary seasonally, providing diverse proxy records of air temperature, sea ice extent and volcanic events (Curran et al., 1998; Palmer et al., 2001; Plummer et al., 2012).

In polar regions, near-surface snow physical properties (e.g., stratigraphy, density) are altered by weather-related processes such as winds, precipitation and temperature fluctuations, producing features at the snow surface and preserving them in the deeper ice as it is progressively buried. These features preserved in ice cores provide crucial and unique records about the past atmospheric variability, offering the possibility of increasing our knowledge of the climate system and potentially allowing for improved predictions of future climate change (Porter and Mosley-Thompson, 2014; Vance et al., 2016). Compared to the 'traditional' climate records from ice cores (gases, isotopes and trace chemistry), these physical features have been relatively poorly studied. Consequently, interpretation of these physical features may be complementary to the traditional chemical interpretation of ice cores, and may help us to understand important aspects of past climate (Jouzel, 2001; Fegyveresi et al., 2018; Orsi et al., 2015).

Air bubbles are common in polar glacier ice. When firn is compressed into ice, bubbles make up about 10% of the ice volume (Cuffey and Paterson, 2010). Bubble density also contains paleoclimate information (Bendel et al., 2013). Transparent ice layers without bubbles have been documented previously in ice cores from both Antarctica and Greenland (Table 1). According to their properties and formation mechanism, these bubble free layers (BFLs) can be classified into three types:

The first type of BFLs are bubble-free bands. Bubble-free bands have been observed in ice cores from a depth range of 200 m to 1,000 m (Faria et al., 2010; Lüthi et al., 2010; Bendel et al., 2013; Uchida et al., 2014). As depth increases, the overburden pressure of snow increases, enhancing the compression of air bubbles between the snow particles. Finally, ice layers will lose all air bubbles, as the gases transform to clathrates. The ice then become completely transparent, forming bubble-free bands, which increases in thickness until at depth all of the ice appears bubble free.

The second type of BFLs are melt layers. Melt layers are used as summer warmth indicators in polar regions (Kameda et al., 1995; Herron et al., 1981). These layers only irregularly appear in ice, and generally form in summer. They are often thick (typically 1–100 mm in thickness) layers with few bubbles and have ragged edges, particularly in shallow cores (Das and Alley, 2005; Orsi et al., 2015).

The third type of BFLs are crusts. Crusts in ice are also characterised by thin ice layers (typically 0.5-1 mm) without bubbles and are found in all seasons on both the Greenland and Antarctic ice sheets (Fegyveresi et al., 2018; Fitzpatrick et al., 2014; Weinhart et al., 2021). Most studies suggest that crusts at the snow surface can be generated via snow metamorphism, controlled by more than one factor, including solar radiation, wind, snowfall, humidity and temperature. Fujii and Kusunoki (1982) measured the sublimation and condensation at the ice sheet surface and suggested that the BFLs may be generated by the condensation of water vapour within the subsurface. Further, based on the observation of glazed surface formation, Albert



**Table 1.** Description of three types of bubble-free layers.

| Type | Thickness | Depth | Cause | Example references |
|------|-----------|-------|-------|--------------------|
| Bubble-free cloudy bands | millimetre to centimetre thick | In air-bubble to clathrate-hydrate transformation zone (200 to 1000 m) | Pressure | Bendel et al. (2013); Faria et al. (2010); Uchida et al. (2014); Lüthi et al. (2010) |
| Melt layer | 1–100 mm | More visible in shallow / firn cores and more prevalent in cores subject to high summer temperatures, e.g. at lower elevations | Summer melt | Alley and Anandakrishnan (1995); Das and Alley (2008); Orsi et al. (2015); Winski et al. (2018) |
| Crusts | 0.5–1mm, up to 2mm | Occur at any depth | Snow metamorphism | Fujii and Kusunoki (1982); Albert et al. (2004); Fegyveresi et al. (2018); Fitzpatrick et al. (2014); Weinhart et al. (2021) |

et al. (2004) argue that the persistent and strong katabatic winds causing wind scour on the East Antarctic plateau enhance the formation of BFLs by increasing vapour transport and air ventilation in firn. During the austral summers from 2008 to

2013, Fegyveresi et al. (2018) measured wind, humidity, temperature and insolation at the West Antarctic Ice Sheet (WAIS) Divide site and suggested that the frequency and thickness of BFLs in the WAIS deep ice core could be used as a proxy for the occurrence of temperature inversions which can enhance firn metamorphism (Fitzpatrick et al., 2014; Fegyveresi et al., 2018). In addition, the crusts from three different Greenland and Antarctic sites were analysed by Weinhart et al. (2021). They suggest that it is difficult to ascribe a single mechanism for all crust formation, as a number of environmental conditions (solar

radiation, humidity, wind, temperature, etc.) could control crust formation in polar snow. However they did find a positive correlation between crusts per annual layer and the (log-transformed) accumulation rate. They also suggest that the BFLs could be used as a summer marker in Antarctica and as seasonal transitions in Greenland, to support ice core dating, owing to their predominantly summertime seasonality at these Antarctic sites and late summer to autumn seasonality in Greenland.

Law Dome is an ice cap on the coast of East Antarctica, with a climate impacted by large weather systems generated in

the Southern Ocean (Bromwich, 1988; Udy et al., 2021; Jong et al., 2022). Thus, the ice at Law Dome broadly preserves the climate signal of East Antarctica, as well as the Indian Ocean and southwest Pacific Oceans. The Dome Summit South (DSS) ice core site (112.81° E, 66.77° S, Fig. 1) is located approximately 4.7 km south-southwest of the Law Dome summit at an elevation of 1,377 m (Roberts et al., 2017; van Ommen and Morgan, 1997). The DSS ice core site is a dry snow zone with a low mean surface temperature (-21.8 °C), relatively moderate wind speeds ($\sim$ 8.3 m $s^{-1}$) and a high accumulation rate of

$\sim$0.680 m $yr^{-1}$ Ice Equivalent (IE) (Morgan and van Ommen, 1997; Crockart et al., 2021).





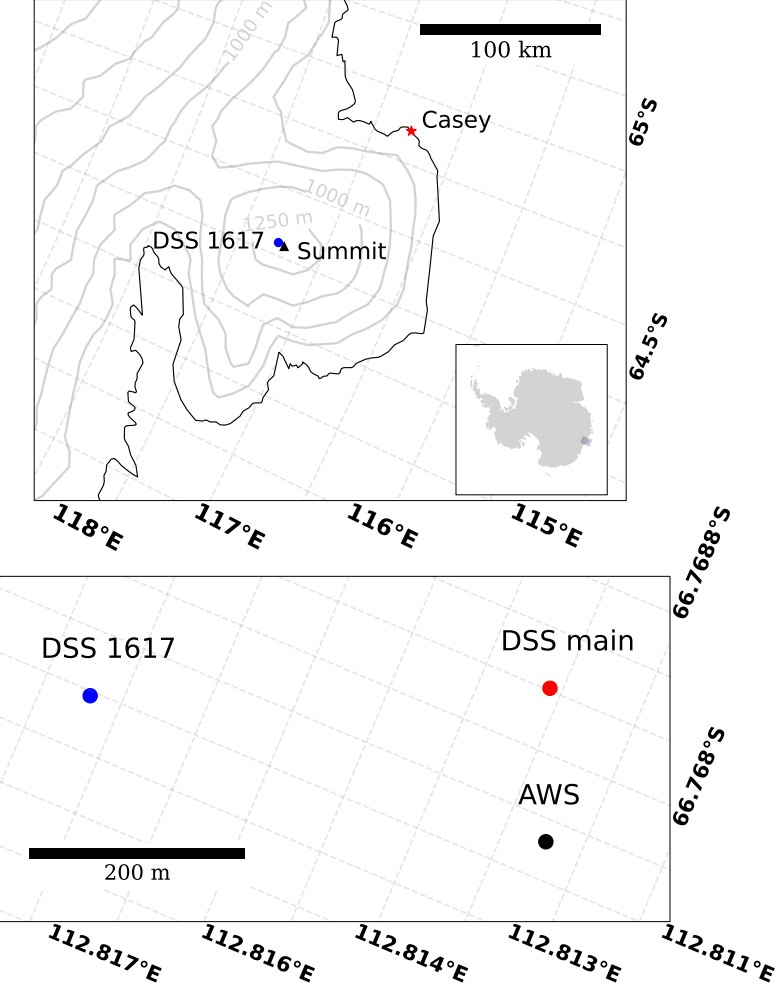

**Figure 1.** The location of the DSS1617 ice core site in relation to the DSS Main core and the automatic weather station which was installed between 1997 and 2003. A snow surface height sensor operated between late 1997 and 2001 inclusive.

The DSS1617 ice core spans most of the recent satellite era, 1990-2017 (Fig. 1). DSS site cores have been used to help understand long-term climate variability in a large region, including East Antarctica and the Indian and Pacific sectors of the Southern Ocean. Upon physical inspection of the DSS1617 core, BFLs were found to occur in almost all cores. The BFLs in DSS1617 are about 1–2 mm thick, and well-defined compared to the surrounding consolidated snow or firn.

There are currently limited observations and understanding of BFL formation. Only a few studies of BFLs in the Antarctic exist, and they focus on the observation of snow surface conditions during snow surface crust formation (Albert et al., 2004; Orsi et al., 2015; Fegyveresi et al., 2018), the links between BFLs and local annual accumulation and the implications of BFL formation on the stable isotope composition of the ice (Dadic et al., 2015). However, the exact formation mechanism of BFLs



and the atmospheric processes related to BFL formation are still not clear. Previous studies have generally been undertaken during summer field observations over short time periods, or are short term studies of 3-5 year long ice core records. This means there is a current lack of the analysis of BFL formation and its relationship to atmospheric processes over time frames long enough to statistically constrain the results.

Since this is an initial study about a new physical feature in East Antarctic glacial ice, this study is both descriptive and exploratory, under the study framework proposed by Leek and Peng (2015). For the descriptive component, we present the BFL record from DSS1617, including the seasonality and annual time series of BFLs. Longer ice cores from DSS have not been routinely scanned for visual stratigraphy using ILCS, making this is the first study of its kind at this site. For the exploratory component, we conduct correlation analyses of BFL occurrence with various atmospheric processes known to drive snow metamorphism, including air temperature inversion, wind scour, accumulation hiatus, high accumulation events and large scale atmospheric circulation. While simplistic in its approach to the exploratory component, this study aims to increase understanding about BFL formation at Law Dome by exploring possible links between BFLs and atmospheric processes.

## 2 Methods

In this study, we used the recently obtained ice core, DSS1617, which was drilled in February 2017 using an electromechanical Eclipse drill. The DSS1617 ice core is 30 metres long from surface and spans the period 1990 to 2016 (Crockart et al., 2021; Jong et al., 2022). All of the DSS1617 ice cores were scanned and analysed for trace chemistry and water stable isotope ratios (in section 2.2). In addition, we scanned the cores using an ILCS (in section 2.1). Climate datasets used in this study include atmospheric reanalysis product ERA5, data from a co-located Automatic Weather station (AWS) at the DSS site, and time series of daily dominant synoptic types in the southern Indian Ocean(Udy et al., 2021). More information about data sets can be found in Table 1 and in the following sections.

### 2.1   Ice core scanned images

The visual stratigraphy is the most fundamental information which can be directly observed from ice cores (Svensson et al., 2005). From the ice layers, the texture, colour, micro-inclusions (e.g. dust particles, salt and other particles), and estimates of size and density of bubbles can be deduced (Alley et al., 1997; Svensson et al., 2005; Weikusat et al., 2017). Researchers can date ice cores by counting visible layers in some cases (primarily Greenlandic cores) and can identify regional to global climate events (e.g. summer melt, volcanic ash layers). In the majority of previous studies, the ice core visual stratigraphy profiles are recorded by drawings or photographs with limited dynamical range and poor resolution. In recent years, ILCS has been developed, with instruments specific to the scanning of whole ice cores developed by Schäfter+Kirchhoff (Germany). These ILCS instruments are designed for the visualization of the laminar structure along the ice core by producing high-quality visual stratigraphic images (Anja et al., 2015). Weikusat et al. (2017) used ILCS to scan a 2,774-meter-long ice core drilled at Kohnen Station, Antarctica (75°00'S, 00°04'E). Krischke et al. (2015) also used ILCS to scan polar ice cores from the Antarctic Ice Sheet (European Project for Ice Coring in Antarctica, EPICA) and Greenland (North Greenland Eemian Ice



**Table 2.** Information about datasets used in this study.

| Dataset | Time period | Location | Elevation | Temporal resolution | Variables |
|---|---|---|---|---|---|
| DSS1617 | 1990 - 2016 | 66°46'26.1" S, 112°48'41.82" E | 1,370 m | Annual to seasonal | Isotopic and chemical record; Annual snow accumulation; Scan image |
| AWS | 1998 - 2001 | 66°46'09" S, 112°48'38" E | 1,376m | Hourly | Snow accumulation; Wind direction at 4m; Wind speed at 4m |
| ERA-5 | 1979 - now | 66.5° S, 112.5° E | - | Hourly | Snowfall; Skin temperature; 2 m air temperature; Wind at 10 m height |
| Synoptic Types (ERA-Interim) | 1979 - 2018 | 30°-75° S, 40°-180° E | 500 hPa | 6-hourly | Geopotential height anomalies |

Drilling Project, NEEM), and used the stratigraphic scans to identify global climatic events, climate induced precipitation and accumulation variations, and date ice cores through counting of annual visible layers where possible. In this study, we use high resolution scans of the DSS1617 ice cores using ILCS to identify BFLs.

The physical features of the DSS1617 ice cores were imaged by ILCS (e.g. the DSS1617 ice core image shown in Fig.2). The ILCS is designed to perform a transect scan of a planar ice core, providing an intermediate layer scan of the ice core for subsequent image analysis. Typically, the ice sample is cut as an ice slab, which is about 1 m long, 100 mm wide and 40 mm thick. For the data described in this study, we scanned the remaining archive half of ice cores previously sectioned for other trace chemical and isotopic analyses. The ice core surface to be imaged was planed using a mechanical thicknesser or planer, with additional polishing by hand using a stainless-steel microtome blade when necessary. The polished slab was then
carefully transferred to the ICLS core carriage, which was then transferred to the ILCS base frame. The ice core slab was imaged a number of times using a range of light and depth settings to ensure good image quality regardless of ice core depth or transparency. DSS1617 was scanned as part of a larger campaign to image multiple ice core archives from East Antarctica via an ILCS instrument on loan from the Canadian Ice Core Lab to the Institute for Marine and Antarctic Sciences (IMAS) in Hobart, Tasmania. Full methods of the scanning protocols developed for this IMAS campaign are under preparation for
publication elsewhere.

Ice samples were imaged by a computer-controlled line scan camera at a resolution of 2,048 pixels. The transmitted light under the ice sample moves synchronously with a camera above the ice, along the core axis (Fig. 3). As the light is focused in the ice slab at a 45° angle relative to the ice surface, the line scan camera can not capture light until impurities or bubbles





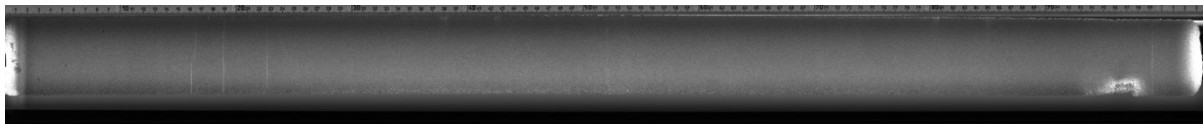

**Figure 2.** An example ice core scan of the DSS1617_24 ice core. The top of the core is at the left of the image, and the core is approximately 105 cm long. A ruler can be faintly discerned at the top of image. Faint vertical white BFLs of about 1 mm thick can be discerned located at 16.5 cm, 19 cm, 22.6 cm, 52 cm and 99.1 cm). The full image gallery of high resolution scans will be provided upon publication.

scatter light into the camera. Because light passes through transparent ice, in deep ice cores below bubble close-off with reduced
scattering, BFLs are recorded as dark lines in the ice core scanned image. However, in order to scan firn cores such as those used in this study, much higher brightness settings were required to obtain adequate images. As a result, the brightness in the scanned image also increases such that the only clear ice in the firn cores (the BFLs) appear as white, rather than dark lines. In this study, we compared the white BFLs in the ILCS images with visual inspection of the ice cores themselves under normal laboratory lighting to confirm that what we interpreted as BFLs were indeed narrow bands of bubble free ice. In addition, due
to limited isostatic pressure near the surface of the ice sheet, the boundary of the DSS1617 BFLs may not be as smooth as BFLs observed in deep ice. This may also contribute to the white appearance of BFLs in shallow ice cores, as the rough edges increase light scattering.

All BFLs were identified from visual inspection of the high resolution images produced using the ILCS. Core drilling logbooks, laboratory processing log books and logbooks kept during the scanning process were consulted to help distinguish
BFLs and core breaks that occasionally occurred during processing, and to double check the presence and depth of BFLs, as many of the BFLs were observed and recorded during core processing. The location of each BFL relative to the depth of each core section was recorded as a dataset of DSS1617 BFL depths for subsequent analysis. This depth dataset was then mapped to year or occurrence using the DSS depth-by-age scale detailed in Jong et al. (2022).

## 2.2 Ice core chemical and isotopic records

The DSS1617 ice cores were sampled at 50 mm resolution for isotope and chemistry analysis, allowing for seasonal to annual dating at DSS. Water stable isotope records ($\delta^{18}$O and $\delta$D) were measured on melted ice samples using a cavity ring-down spectrometer (Picarro L2130-$i$) following established methods (Curran and Palmer, 2001; Palmer et al., 2001; Curran et al., 2003; Plummer et al., 2012; Jong et al., 2022). Analysis of trace ion chemistry was conducted by ion chromatography (Thermo-Fisher/Dionex ICS3000), which provides the concentrations of ions, including chloride (Cl$^-$), sodium (Na$^+$), magnesium
(Mg$^{2+}$), calcium (Ca$^{2+}$), potassium (K$^+$), sulphate (SO$_4^{2-}$), non-sea-salt sulphate (nssSO$_4^{2-}$), nitrate (NO$_3$) and methane sulfonic acid (MSA$^-$). Analytical methods were mostly based on Plummer et al. (2012), except for a change in the cation analytical column to an Ionpac CS19 to improve detection and peak resolution of Mg$^{2+}$ and Ca$^{2+}$ (Jong et al., 2022).



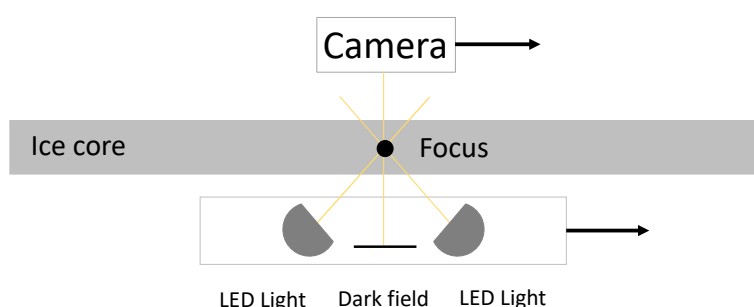

**Figure 3.** Ice core scanner process schematic. The arrows indicate the direction of travel of the camera and light source along the plane of the core from top to bottom

## 2.3 Ice core dating

The DSS1617 ice core was dated using annual layer counting. The calendar year boundaries in the ice core (Fig.4) were
defined according to the seasonal variation in a suite of chemical and isotopic species, with the stable water isotope record as the primary seasonal indicator (Morgan and van Ommen, 1997; Jong et al., 2022). The summer can be indicated by the peak of hydrogen peroxide ($H_2O_2$), $nssSO_4^{2-}$, MSA and the sulphate/chloride ($SO_4^{2-}$ /$Cl^-$) ratio, while the sea salt species ($Cl^-$, Na+, $Mg^+$) show peaks in winter. In addition, a volcanic reference horizon is present in the DSS1617 core, identified by an increase in sulphate signal beyond background levels ($nssSO_4^{2-}$ peaks) related to the volcanic eruption of Pinatubo, Philippines, in 1991
(Plummer et al., 2012).

In this study, monthly dating was also undertaken on the DSS1617 ice core. Recent advances in analytical technologies have enabled ice core researchers to analyse polar ice cores at a high temporal resolution on annual to seasonal time scales (Steig et al., 2005; Iizuka et al., 2006; Hoshina et al., 2016). Fegyveresi et al. (2018) tried to determine the monthly boundaries in an ice core by equally dividing accumulation between annual boundaries into 12 parts. However, precipitation in East Antarctica
can be episodic and seasonally biased (Crockart et al., 2021). Thus, monthly boundaries produced by dividing an ice core annual layer into 12 equal parts may not be accurate enough for very high resolution studies such as this one. A more accurate dating method is required to enable the detailed investigation of BFL formation, preservation and atmospheric drivers for this study. In recent years, atmospheric reanalysis data have been widely used for various studies such as surface climate characteristics of the Antarctic region as well as large-scale atmospheric forcing mechanisms modifying the surface climate variables (Hines et al.,
2000; Maksym and Markus, 2008; Lenaerts et al., 2012; Costi et al., 2018). ERA5 is the fifth generation atmospheric reanalysis product of the global climate produced by the Copernicus Climate Change Service (C3S) at ECMWF (Hersbach et al., 2019). According to the evaluation of near surface atmospheric variables in the reanalysis products over Antarctica, ERA-5 performs



better than previous reanalysis products on most of the variables used here, especially temperature and snowfall (Dutra et al., 2010; Berrisford et al., 2011; Tastula et al., 2013; Balsamo et al., 2018; Hoffmann et al., 2019; Gossart et al., 2019; Hersbach et al., 2020). We test the ability of ERA5 to represent local precipitation variability at the DSS1617 ice core site by comparing 1). the ice-core-derived and ERA5-derived annual accumulation between 1990-2016, and 2). the AWS snow surface height and ERA5 cumulative precipitation between 1998-2001. The significant correlation (r=0.907, p=0.01) between the ice-core-derived and ERA5-derived annual accumulation supports our use of ERA5 in ice core annual dating. The comparison between AWS snow surface height and ERA5 cumulative precipitation shows that ERA5 can present the trend of snow surface increase well (Fig.A1), which give us confidence that ERA5 can be used for monthly ice core dating.

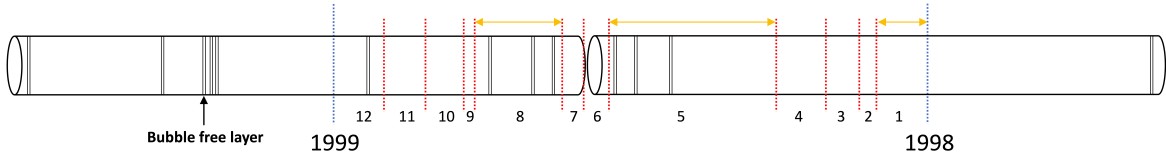

**Figure 4.** Schematic describing the estimation of a monthly dating scale for DSS1617 to facilitate the high resolution dating of observed BFLs. The blue lines are the annual year horizons for DSS1617 from Jong et al. (2022) (determined by using the established understanding of seasonal variation of chemistry species at DSS). The red lines are the boundary layers of months (numbered 1 to 12, derived from monthly dating by using ERA5 to proportion the annual accumulation into 12 unequal sections). The annual boundary of 1998 is also the beginning of January. The first red line on the left in 1998 is the end of January and also the beginning of February. The yellow lines are the accumulation in January, May and August 1998, clearly demonstrating the episodic nature of monthly snowfall in any given year.

We calculated ERA5-derived monthly accumulation by summing of ERA5 hourly snowfall within each month from 1990 to 2016. To convert ERA5-derived monthly accumulation into ice core monthly accumulation in any given year, we calculated the ratio of ice-core-derived and ERA5-derived annual accumulation. This ratio accounts for densification of ice down the core. Through this ratio for each given year, we converted the ERA5-derived accumulation for the 12 months of any given year into the proportional monthly accumulation received at the ice core site:

$$Acc_{ic} = Acc_{ERA5} * r \qquad (1)$$

Where $Acc_{ic}$ is the accumulation for each month in the ice core, $Acc_{ERA5}$ is the accumulation for each month calculated by ERA5 data (Table1), and $r$ is the ratio of ice-core-derived and ERA5-derived annual accumulation for each year.

Then, based on this monthly ice core accumulation, approximate monthly boundaries were produced by adding up the monthly accumulations sequentially from the beginning of each year (Fig.4). Due to inherent error in this method (primarily that ERA-5 accurately represents accumulation at the DSS site on monthly scales), the confidence in monthly dating is lower than for the annual dating derived from analysis of seasonally varying chemical and isotopic species.



## 2.4 Estimating ice core BFL position from accumulation data measured via a co-located Automatic Weather Station

For a subset of the DSS1617 record, we can also map accumulation using locally-acquired accumulation data. Hourly snow
accumulation data (snow surface height) was measured via a co-located AWS at the DSS site between 1997-2002 (Dome
Summit South Automatic Weather Station; date of installation: December 20,1997) from the Australian Antarctic Division
(AAD) network (http://aws.cdaso.cloud.edu.au/). The DSS AWS was equipped with a snow surface height sensor, which al-
lowed identification of individual precipitation events. The average annual accumulation for DSS1617 core is 0.69 m (IE).
This sufficiently high accumulation rate together with accurate local snow accumulation data allows us to date the BFLs at a
sub-monthly scale for this short subset of the DSS1617 record. Thus, we attempt to estimate the date of BFL formation (or
preservation) during this period from the AWS data.

From the annual layer horizons of the DSS1617 chemistry record as defined in Jong et al. (2022), the depths of each calendar
year boundaries are available. We mapped these year horizons onto the scanned images of the DSS1617 cores using the core
number and length measurements of each core. In each year, the location of every BFL is identified in the ILCS images. We
calculated the distance of each BFL from the prior year horizon, then transcribed this to the fraction of the net snowfall surface
height gain for that year from the AWS data by assigning 1 January to 0% and 31 December to 100%. For example, if a BFL
was located at 25% of the total accumulation for the year it occurred in, we determined this BFL to occur when the AWS
surface height increased by 25% of the total surface height increase for that year. Because the AWS data is observed snow
surface height variation, it is subject to both accumulation and redistribution/erosion under the sensor. This means there is
rarely a steady (if episodic) increase; rather, the surface height is dynamic, with small increases occurring alongside negative
surface height changes as snow is redistributed or eroded. Thus, the time when accumulation reaches a specific fraction (such
as 25%) could occur at more than one point in time. We have therefore applied a time uncertainty (error boundary) for each
BFL identified during the 1997-2002 period when snow surface height was measured locally by AWS. This error boundary
defines the first and last time the surface height sensor data reaches the fraction of the net snowfall surface height gain for that
year before each BFL. Finally, within this time error boundary, we placed the BFL at the date where the snow surface height
does not return below a clear erosion boundary, as we presume that the establishment of an erosion boundary likely indicates
a crust or BFL formation rather than loose surface snow. This is our estimated BFL formation date.

## 2.5 Links to atmospheric processes

As mentioned in section 1, the relationship between BFLs and climate in previous studies has mainly been detected by sum-
mertime field observations of snow surface crust formation, lacking full-season, long-term research. Here, we make an attempt
to detect the relationship between atmospheric processes and DSS1617 BFLs over the past 27 years. Long-term meteorological
records for 1990-2016 at the ice core sites are therefore required.

Time series of daily synoptic types from Udy et al. (2021), which characterized the daily dominant synoptic types in the
southern Indian Ocean for 1979-2018 by using self-organising maps are used. In this dataset, similar circulation patterns (500-
hPa geopotential height anomalies) were grouped into nine synoptic types. During 1979-2018, each day is assigned one of the





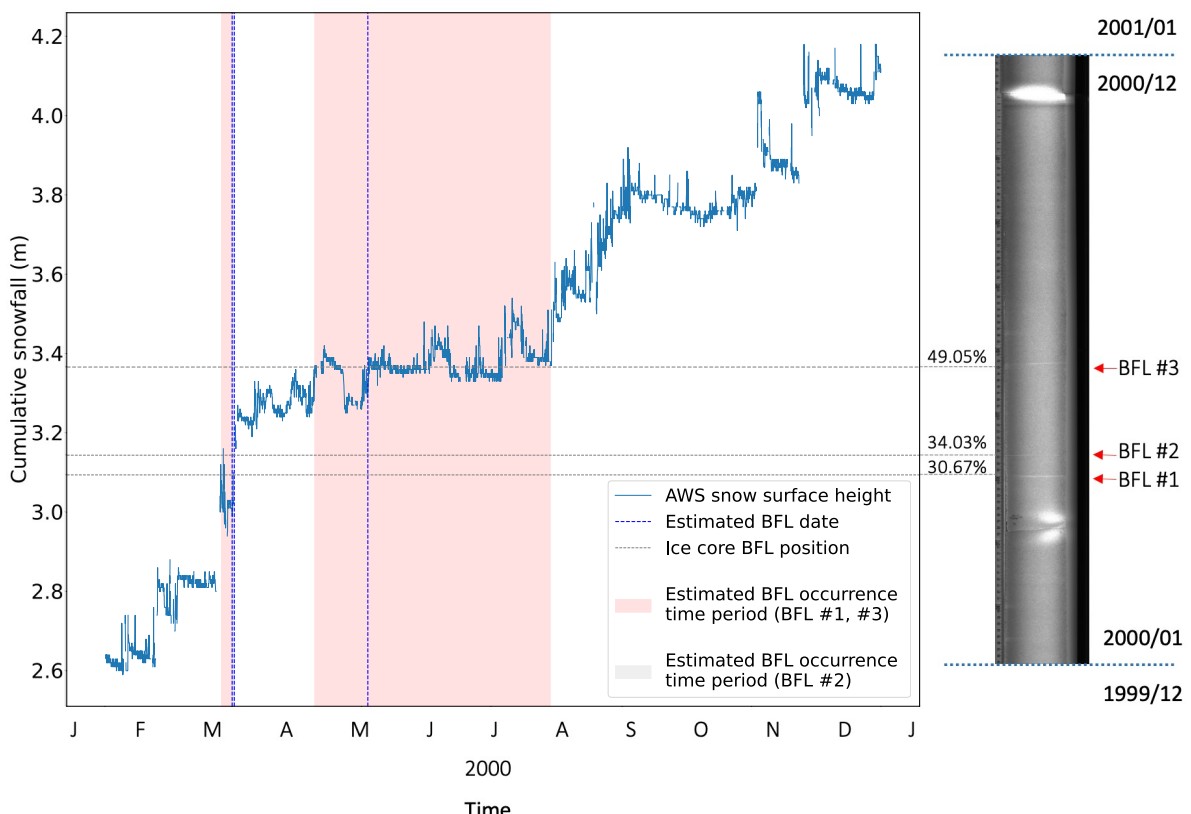

**Figure 5.** Example of aligning AWS dates of snowfall surface height changes to BFLs in the DSS1617 ice cores. Two ice core ILCS images are shown (DSS1617-20 and DSS1617-21). The images have been cropped to include only ice within the 1999/2000 and 2000/2001 year horizon/boundaries (dotted blue horizontal line). This includes three BFLs in total, indicated by the red arrows. We then mapped these BFLs identified in the ILCS images to the calculated fraction of the cumulative snow surface height (blue line) for the year 2000 as measured by the AWS snow surface height sensor, where the difference between snow surface height in 1 January and 31 December is the AWS annual accumulation in the year 2000 (dotted black horizontal lines). For example, BFL #1 for the year 2000 occurred at 30.67 % of the annual accumulation (approximately March). The red shading indicates the first and last time when the snow surface height reached this snowfall level (to within 1 cm) to define the time period where BFL formation likely occurred. The dotted blue vertical line indicates the point at which the snow surface height appeared restricted from height losses (i.e., there was an erosion boundary).

nine synoptic types. Time series of synoptic types provide daily large-scale synoptic patterns over the southern Indian Ocean adjacent to the DSS site, which are then used to investigate the relationship between large-scale atmospheric circulation and DSS1617 BFLs.

In addition to large-scale atmospheric circulation, local atmospheric processes may also be related to BFLs, including
surface-based temperature inversion, wind scour, accumulation hiatus and high precipitation events. As no dataset of these





variables exists, we use climate variables from ERA5 (Table 2) from 1990-2016 to detect their date of occurrence and then produce such time series.

We selected atmospheric process events by setting an exceedance threshold. First, we conducted a sensitivity test, choosing the threshold which resulted in the highest correlation coefficient between the number of BFLs per year (and per month) and

the number of atmospheric processes per year. This gives the highest possible correlation between BFL and each atmospheric process. Then, this choice of threshold was subjectively assessed to ensure physically realistic values were chosen. Unrealistic thresholds were removed, and the threshold resulting in the next-highest correlation coefficient was subjectively assessed. This process was repeated until the threshold with a physically realistic value giving the highest correlation coefficient was chosen (Table. 3).

A surface-based temperature inversion is the phenomenon in which the air above the snow surface is warmer than the air temperature at the snow surface, which can lead to upward vapour transport between snow condense at snow surface and form crust. In general, this can be detected by measuring air temperatures at different heights. Adolph et al. (2018) suggest that the difference between skin temperature and 2 m temperature from satellite remote sensing have the potential to be used to detect the existence of temperature inversions. Here, we used hourly 2 m temperature air temperature and skin temperature from

ERA5, calculating the temperature difference between them to obtain inversion strength (TIS):

$$TIS = T_{2m} - T_{skin} \tag{2}$$

where $T_{2m}$ is the 2 m air temperature and $T_{skin}$ is the skin temperature. Here, a temperature inversion is defined as $T_{2m} > T_{skin}$, or $TIS > 0$. Since BFLs only appear between 0-7 times a year, we need to detect the events that happen only a limited number of times per year. Here, we select atmospheric process events by setting an exceedance threshold. In this case, the two

variables important for temperature inversion events are TIS and the duration ($D_{TI}$). We test a series of thresholds for TIS (from 0 to 10 K) and $D_{TI}$ (from 1 to 24 h), to filter temperature inversion events which are sufficiently strong in magnitude and duration. Based on different combinations of TIS and $D_{TI}$, we produce the corresponding annual time series of exceedances, and then compare the annual total time series of temperature inversion exceedances with the annual total time series of BFLs. There are several TIS and $D_{TI}$ thresholds combinations that can produce the same number of annual temperature inversion

events occurrence at the same level as BFLs. Among them, the combination of thresholds based on the annual temperature inversion time series most related to BFLs was chosen as the final threshold (tis and $d_{TI}$). Seasonal and annual time series are produced for correlation tests with BFLs.

An accumulation hiatus is defined as a period of time with little to no snow accumulation (Courville et al., 2007). Thus, the main variables for accumulation hiatus event detection are daily accumulation (A) and hiatus duration ($D_{AH}$). A is calculated

by summation of ERA5 hourly snowfall data in each day. As with the threshold tests before, the threshold $D_{AH}$ ($d_{ah}$) and threshold A ($a_{ah}$) are determined from a series of sensitivity studies ($D_{AH}$: 0 to 20 days; A: 0 to 0.6 m) to detect accumulation hiatus event with unusually low daily accumulation and long hiatus duration. Seasonal and annual time series were also produced for correlation tests with BFLs.



Daily accumulation (A) and duration ($D_{HP}$) are also important for characterizing high precipitation events. Using similar
tests as for accumulation hiatuses, the threshold duration ($d_{hp}$) and threshold precipitation ($a_{hp}$) are determined to detect high
precipitation event by satisfying $D_{HP} > d_{hp}$ and A > $a_{hp}$. Seasonal and annual time series for high precipitation events were
produced for correlation tests with BFLs.

Wind scour occurs where strong katabatic winds persistently polish the ice surface (Fisher et al., 1983). Thus, the main
quantities of interest in accumulation hiatus events are wind speed and wind persistence. The 10 m u-component of wind and
10 m v-component of wind in ERA5 are used here to calculate the 6 h average wind speed (S) and 6 h average wind persistence
($P$). P is a calculated wind persistence metric (Fraser et al., 2016) defined as:

$$P = \frac{\sqrt{U^2 + V^2}}{S} \tag{3}$$

where U is the mean zonal wind component in 6 h, V is the mean meridional wind component in 6 h, and S is the mean wind
speed in 6 h. $P$ gives a number between 0 to 1 with a value of 1 indicating that wind always blows from the same direction.
Here, we test a series of thresholds for S (0-20 m $s^{-1}$ in increments of 0.1) and $P$ (0 0.9999 in increments of 0.0001). As with
previous atmospheric exceedances, we select the mean wind speed thresholds (s) and wind persistence thresholds (p) that can
reduce the number of exceedances to about 5 per year, and result in the highest correlation between annual/seasonal total wind
scour exceedances and BFL. Based on this, seasonal and annual time series were generated for correlation tests with BFLs.

To explore the links between atmospheric processes and BFLs, we tested whether a statistically significant correlation
between the time series exists. As mentioned previously, we produced annual time series and seasonal (three-monthly) time
series for all atmospheric processes and BFLs, and then calculated the correlation coefficients. The correlations were considered
significant at a p-value of $\leq 0.05$.

Although we can produce monthly time series for both BFLs and atmospheric processes, we only produce the correlation
between time series at annual and seasonal time scales, due to relatively limited confidence in monthly dating. The way we date
BFLs relies on snow accumulation which can only record the time when the BFLs are buried by snowfall. Thus, the BFLs could
have formed during precipitation, or in any time between two high snowfall events, but are only recorded by the precipitation
after formation. For example, if a BFL formed in December followed by an accumulation hiatus of weeks (and up to month),
this BFL would be preserved by a snowfall event in January. In this case, the monthly time series would record it as a BFL in
January rather than December. However, in seasonal time series BFLs would occur in summer, reducing this type of error. The
DSS1617 ice core site is a high accumulation site, therefore we can know accumulation hiatuses lasting months in duration are
rare. Thus, in this study, seasonal time series are sufficient for reducing the dating error caused by accumulation hiatuses.



# 3 Results

## 3.1 Annual and seasonal frequency of bubble-free layer occurrence

A total of 139 BFLs were identified in the DSS1617 ice cores during the 1990-2017 period, with a range of 0 to 9 $y^{-1}$
(mean 5.148 $y^{-1}$). No BFLs occurred in 2014, while they appeared most frequently in 2008 (nine BFLs; Fig.6a). BFLs
occurred year-round, but were most commonly observed in austral autumn and winter, specifically April to July (Fig.6b). The
time series of annual total BFLs shows a marginally significant decreasing trend since 1990 (slope=-0.127 BFLs/year; 95%
confidence interval=[-0.220,-0.034]). However, this decreasing trend is likely to be largely influenced by 2014 (which recorded
no BFLs). This decreasing trend is largely influenced by 2014 (which recorded no BFLs). Analysis of a shortened record from
1990 to 2013 indicates that this trend is insignificant (slope=-0.0961, BFLs/year; 95% confidence interval=[-0.201, 0.009]).

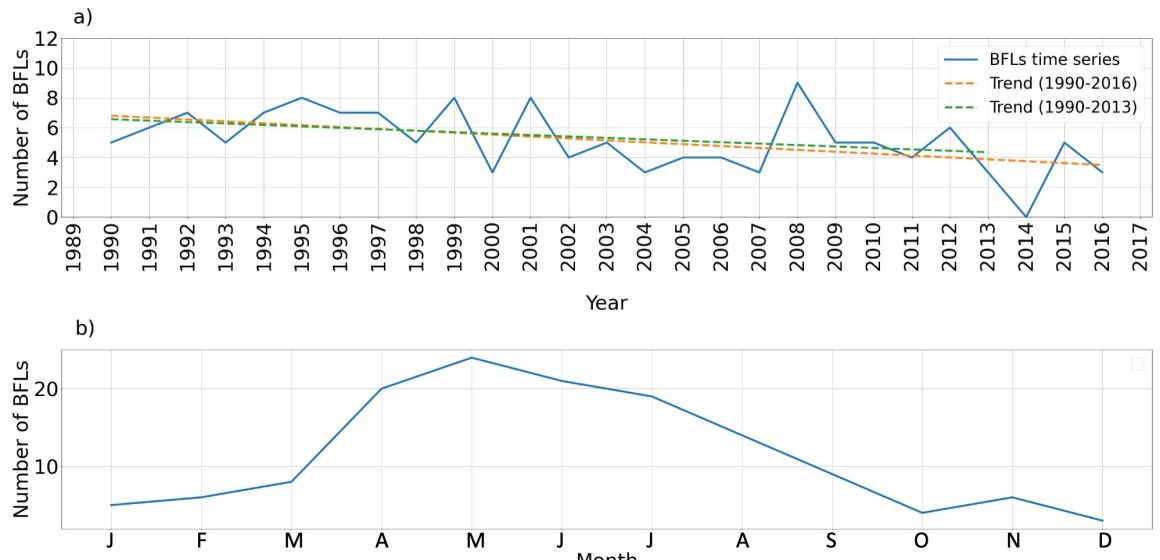

**Figure 6.** Time series of number of bubble-free layers between 1990 to 2017 (a), and the monthly frequency of bubble-free layers in the
DSS1617 ice core (b).

## 3.2 Investigation of bubble free layer occurrence using a local automatic weather station

We investigated where BFLs occurred temporally in the ice core by comparison with the local AWS data compiled by McMor-
row et al. (2004). Blue bars in Figure 7 show the occurrence of BFLs in comparison to local snowfall accumulation measured
by the AWS at Law Dome Summit. It is important to note that the blue bars indicating a BFL are unlikely to denote the exact
time of BFL formation in the snowpack when BFLs are preceded by a period of no snowfall for days or weeks (an accumulation
hiatus). This is because BFL formation could occur at anytime during the accumulation hiatus, as the ice core does not record
temporal information during accumulation hiatuses.





Three BFLs occurred in 1999 that we found difficult to estimate occurrence dates for (marked by grey lines). This was because of an extended period of very low accumulation during June to December 1999. Qualitatively, most BFLs appear to
occur when snow accumulates rapidly in a short period of time during high precipitation events. This can be seen as a period where the snow surface height increases significantly over hours to days (Fig. 7). As previously noted, these high accumulation events often occur after a long period of days or weeks without any accumulation, which we term an accumulation hiatus. Furthermore, there are also times when this accumulation hiatus or high precipitation event scenario is recorded by the AWS and we would expect the occurrence of a BFL, but there is no evidence in the DSS1617 core of one (e.g., March 1999 and
November 2001).

### 3.3 Annual and seasonal time series of atmospheric processes

We compared the time series for atmospheric processes and BFLs at both annual and seasonal scales, to investigate the relationship between regional weather variability and the occurrence of BFLs. The results of comparisons of the BFLs with atmospheric processes are shown in Table 3. Accumulation hiatuses show the most significant (negative) correlation with BFLs at both sea-
sonal and annual scales. Temperature inversions, wind scour and high precipitation events are also significantly correlated with BFLs, but only at the annual scale. However, as noted in the methods, we performed numerous correlation analyses for multiple thresholds of wind scour. Depending on the threshold, we were able to obtain marginally significant correlations at either the seasonal or annual scale, but the threshold required to achieve marginal significance needed a far greater number of wind scour events per year than the number of BFLs per year. As a result, we don't consider the wind scour results to be as robust as the
results for the other three processes as we cannot discount that these may be spurious correlations related to the high number of wind scour events.

### 3.4 Annual and seasonal time series of synoptic types

We compared the time series for synoptic types based on self-organising maps (SOMs) from Udy et al. (2021) and BFLs at both annual and seasonal scales, to investigate the relationship between large scale synoptic variability and the occurrence
of BFLs. The results of comparisons of the BFLs with synoptic types are shown in Table 2. Most comparisons show an insignificant correlation with BFLs, and all are insignificant at the seasonal scale. Only synoptic types 1 and 2 (SOM1, SOM2) are significantly correlated with BFLs at the annual scale (positively for SOM1 and negatively for SOM2).

## 4 Discussion

### 4.1 Annual and Seasonal frequency of bubble-free layer occurrence

The time series of annual total BFLs shows a significant decreasing trend since 1990 (slope=-0.127 BFLs/year; 95% confidence interval=[-0.220,-0.034]). However, we tested whether this significance is merely due to the lack of BFLs in 2014 (number of BFLs in 2014 is 0) by performing trend analysis over the time period 1990 to 2013. Over this time period, the trend drops



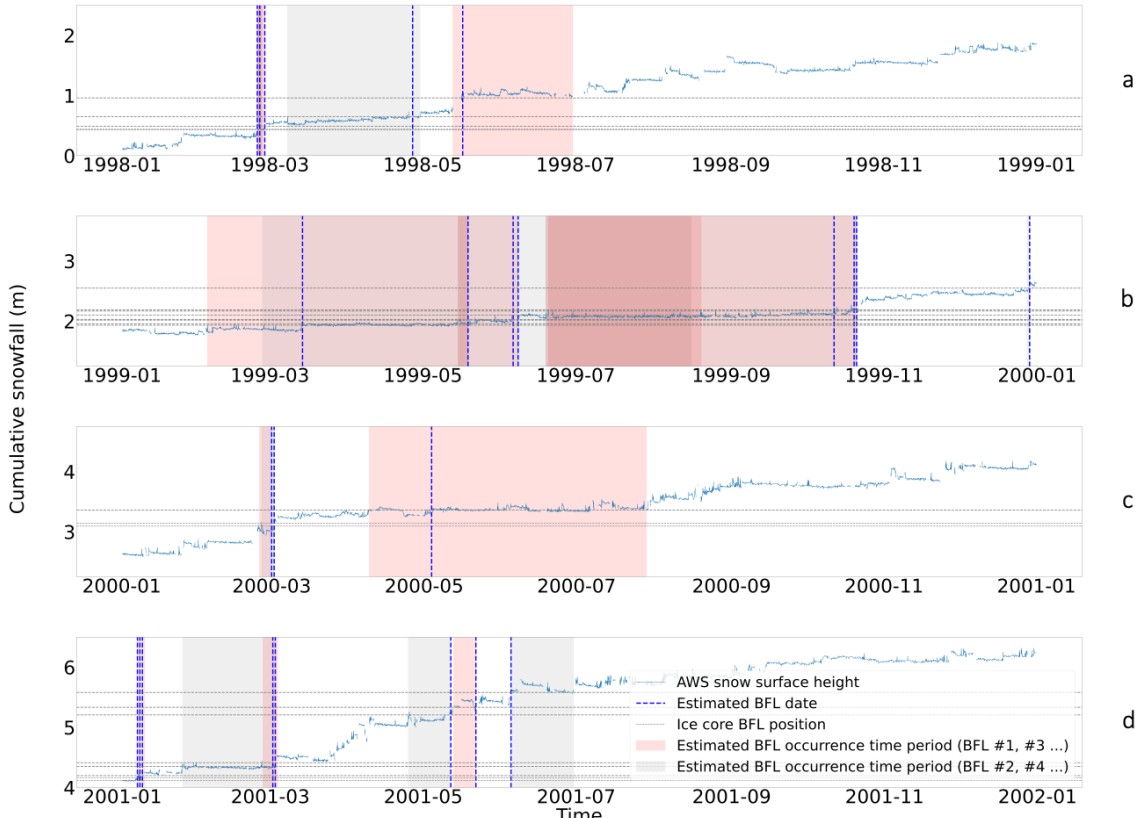

**Figure 7.** Cumulative snowfall (see McMorrow et al. (2004) and BFLs. Hourly snowfall from the Automatic Weather station (AWS) at Law Dome Summit, approximately 500 m southeast from the DSS1617 drill site (blue line). Horizontal dashed lines indicate the fraction of total accumulation in metres water equivalent (m) where BFLs were identified in the ice cores, based on the annual layer horizons of Jong et al. (2022). Dashed red vertical lines indicate the estimated formation date of the BFLs based on the criteria defined in Section 2.4. Pink and grey banding around each estimated BFL formation date indicate the qualitative boundaries for each BFLs, as based on the criteria in Section 2.4. Note different banding colours are merely to differentiate individual BFL zones, with the first BFL for each year being red, second being grey, third being red and so forth.

slightly to be insignificant (slope=-0.0961, BFLs/year; 95% confidence interval=[-0.201, 0.009]). There are numerous caveats that may lead to an artificially significant trend: 1. The surface firn cores may be more fragile, being composed of primarily compressed snow. Core breaks may preferentially occur at BFLs, in which case they would not necessarily be picked up by our visual counting; 2. It is possible that in very near surface firn cores where the refraction in the imaging process is very high, we may not detect by our visual process faint or very thin BFLs. As a result, this trend may be spurious, and we will need to analyse longer data sets with deeper cores in the future to determine if there are longer-term trends in BFL formation.



**Table 3.** Threshold of atmospheric processes and the corresponding correlation coefficient between BFL occurrence and each atmospheric process threshold exceedance. Values in brackets are the thresholds/correlation coefficient values before subjective adjustment. $r_a$ and $p_a$ are the correlation coefficient and p value between annual time series of BFLs and atmospheric processes, respectively. $r_s$ and $p_s$ are the correlation coefficient and p value between seasonal time series of BFLs and atmospheric processes, respectively. Significance is assessed based on two-tailed test at < 0.05. Bold type indicates significant correlations.

| | | | Correlation coefficient | | | |
| | | | Annual | | seasonal | |
| Atmospheric processes | Thresholds | | $r_a$ | $p_a$ | $r_s$ | $p_s$ |
|---|---|---|---|---|---|---|
| **Temperature Inversion** | tis = 8.855 | $D_{TI}$ = 12h | **-0.405** | 0.036 | 0.014 | 0.889 |
| **Wind scour** | p = 0.9999 | s = 14.151 | **-0.548** | 0.003 | 0.132 | 0.176 |
| | | (s = 12.75) | (**-0.57**) | (0.002) | (0.118) | (0.228) |
| **Accumulation hiatus** | $d_{ah}$ = 4 | $a_{ah}$ = 0.002 | **-0.554** | 0.003 | **-0.225** | 0.02 |
| **High precipitation** | $d_{hp}$ = 0 | $a_{hp}$ = 0.023 | **0.397** | 0.04 | 0.152 | 0.117 |

The seasonality of BFLs indicates that at the Law Dome site, the majority of BFLs occur in autumn and winter. This contrasts
to the findings of Fegyveresi et al. (2018) and Weinhart et al. (2021) working at the WAIS Divide and Greenland ice core sites,
respectively, where BFLs predominantly occurred in summer (WAIS) or late summer and autumn (Greenland). This seasonality
suggests that BFLs may be related to regional climate variability, rather than local surface processes. This is important because
it indicates that DSS1617 BFLs have the potential to provide past climate information. Thus, exploring what causes BFLs has
important implications for studying the climate of the Law Dome region. Secondly, the autumn/winter seasonality at the DSS
Law Dome site is strong evidence that these BFLs are not melt layers. As mentioned in Section 1, melt layers usually form in
summer due to both radiation- and temperature-related melting processes, and they often have a different appearance, being
broader and with ragged edges (Das and Alley, 2005; Orsi et al., 2015). The BFLs in this study are narrow and have clearly
defined edges, similar to those identified at a glazed site (80.78 ° S, 124.5 ° E) at East Antarctica and shown to not be due
to melt processes (Albert et al., 2004). Lastly, the Law Dome site is also at an elevation and location that it is not known to
experience regular melt events even at the height of summer (van Ommen and Morgan, 1997; Morgan and van Ommen, 1997),
a finding which is supported by remote sensing-based studies of melt distribution (Trusel et al., 2012).

However, there are aspects to both the previous studies and our own work that mean interpretations of the seasonality
of BFLs are not yet sufficiently advanced to definitively date BFLs, either at Law Dome or other sites. Neither our dating
method nor that of previous authors can currently date these features to the required accuracy (month). Fegyveresi et al. (2018)
assume that the annual horizon occurs on 1 January. They then divide the annual snow accumulation into 12 equal parts, to
derive estimated monthly accumulation, assuming uniform accumulation rates throughout the year. Given our findings around
accumulation hiatuses, high precipitation events and highly variable monthly accumulation rates, this assumption may not hold





**Table 4.** The correlation coefficient BFLs and each synoptic type. $r_a$ and $p_a$ are the correlation coefficient and p value between annual time series of BFLs and atmospheric processes, respectively. $r_s$ and $p_s$ are the correlation coefficient and p value between seasonal time series of BFLs and atmospheric processes, respectively. The significance is assessed based on two-tailed test at < 0.05. Bold type indicates significant processes and correlations.

| | Correlation coefficient | | | |
|---|---|---|---|---|
| Synoptic types | Annual (n=27) | | Seasonal (n=107) | |
| | $r_a$ | $p_a$ | $r_s$ | $p_s$ |
| **SOM1** | **0.383** | 0.048 | 0.01 | 0.86 |
| **SOM2** | **-0.523** | 0.005 | -0.012 | 0.826 |
| SOM3 | -0.141 | 0.482 | 0.083 | 0.136 |
| SOM4 | 0.175 | 0.382 | -0.092 | 0.1 |
| SOM5 | -0.285 | 0.15 | -0.052 | 0.351 |
| SOM6 | 0.16 | 0.427 | 0.106 | 0.057 |
| SOM7 | -0.001 | 0.998 | -0.029 | 0.598 |
| SOM8 | 0.155 | 0.439 | -0.028 | 0.617 |
| SOM9 | -0.041 | 0.839 | -0.009 | 0.875 |

at our site (and may be an over-simplification at the WAIS Divide site). This is an important assumption in prior studies that may need to be revisited. Weinhart et al. (2021) use five clear seasonal cycles in $Cl^-/Na^+$ and $NH_4^+$ in a Greenland ice core 365 (EGRIP, W10) to estimate summertime intervals. In this study, we use ERA5 snowfall data, combined with DSS1617 annual dating from ice core dating (which incorporates understanding of chemistry seasonal cycles at Law Dome), to calculate the monthly accumulation portion at the DSS1617 site (see Appendix A1). The DSS1617 annual dating is considered highly robust (Plummer et al., 2012; Jong et al., 2022). While using reanalysis snowfall to simulate accumulation at the ice core site will still result in dating errors, this approach is likely a more robust approach for studying BFL formation. However, given the above 370 caveats, we analyse BFLs on a seasonal scale rather than a monthly scale in this paper. This coarsening of timescale should reduce any inherent bias resulting from dating errors compared to monthly dating.

In addition, the difference between the monthly dating approach of this and previous studies could lead to a difference in reported BFL seasonality to some extent, but these should be minor. Most DSS1617 BFLs occur during late autumn and winter, a time period with few BFLs in ice cores analysed in previous studies (Fegyveresi et al., 2018; Weinhart et al., 2021). Thus, 375 the DSS1617 BFLs may be caused by different processes than those mentioned in previous studies.





### 4.2 Investigation of BFL occurrence in conjunction with local site AWS snow accumulation data

From AWS records, snow surface height at Law Dome does not increase continuously and uniformly (McMorrow et al., 2001; McMorrow et al., 2004). Not only do accumulation hiatuses and high precipitation events occur relatively frequently, but the snow surface height also often decreases, likely driven by wind removal and erosion (Fig. 5, Fig. 7). Here, we note in several instances that the snow surface height erodes to a certain level of apparent resistance. This resistance level constitutes a dynamic 'barrier', which arrests further erosion. For example, in the AWS accumulation record after the fifth BFL in 1998 (fig.7a) and the third BFL in 2000 (fig.5, fig.7c), the snow surface height becomes eroded back to the same accumulation boundary. From these facts we infer that such barriers could be BFLs.

The DSS1617 ice core preserves an extremely low accumulation year – 1999 – that is unusual both during the satellite era and over recent centuries (Fig. 7 and van Ommen and Morgan, 2010). Without sufficient accumulation, seasonal BFL dating in 1999 becomes challenging. Compared with BFLs in other years, there is a higher than average number of BFLs in 1999, however their appearance is qualitatively different to other years, in that they are crooked, broken and/or not perpendicular to the plane of the core. This may be related to the extremely low annual accumulation in this year (van Ommen and Morgan, 2010). If BFLs in years of extremely low accumulation are unlikely to present as a straight line shape, this may be evidence that sufficient snowfall plays an important role in BFL preservation. If so, while low accumulation years may lead to BFL formation, sufficient accumulation is needed to preserve them in the ice core record. In order to disentangle formation and preservation of BFLs, and the relationship between episodic accumulation (i.e., event scale hiatuses and precipitation) and annual accumulation (yearly total) we will need to investigate other ice core records over the satellite era, and also over longer timescales.

### 4.3 Annual and seasonal time series of atmospheric processes

From Table 3, the number of accumulation hiatuses per year is the only atmospheric process which is significantly correlated with DSS1617 BFL occurrence on both annual and seasonal time-scales ($r_a$=-0.554, $p_a$=0.003; $r_s$=-0.225, $p_s$=0.02). This is strong evidence that accumulation hiatus events are related to the formation of BFLs. This negative correlation indicates that years with more short-term accumulation hiatuses result in fewer BFLs in the DSS1617 ice core. From the AWS records in Section 4.2, we also know that accumulation hiatuses occur frequently at the DSS ice core site (McMorrow et al., 2004). As mentioned in the introduction, this study aims to increase understanding about BFL formation at Law Dome by exploring possible links between BFLs and atmospheric processes. The significant relationship between BFLs and the number of accumulation hiatuses provides direction for future research. In future studies, we plan to concentrate on any links between accumulation hiatuses and BFLs, by exploring BFL formation mechanism/s and the role of accumulation hiatuses using snow-pack models.

From Table 3, the number of temperature inversions, high precipitation events and wind scours per year only show a significant correlation with BFL numbers on an annual scale. These processes were not correlated significantly at both seasonal and annual time-scales, indicating either weaker or more complex relationships with BFL formation than accumulation hiatuses. However, this result does not necessarily mean that these three atmospheric processes are not relevant to BFL formation - while





the relationship is not as strong as that of accumulation hiatuses, it still exists. The four atmospheric processes considered in this
study are all very complex processes, with considerable interplay and inter-dependence. Here, we select atmospheric process
events by setting an exceedance threshold. The result is highly dependent on the values of each threshold and duration of each
atmospheric process event. This leads to the conclusion that it is difficult to accurately detect the relationship between BFLs
and such complex processes using simple, single-variable correlations. That is, this method can only be used to test the rela-
tionship between BFL occurrence and individual atmospheric processes, but the influence of multiple atmospheric processes
on BFLs cannot be effectively explored with this simple method. BFLs may be the result of the combined action of two or more
atmospheric processes, such as our earlier discussion around BFL formation during accumulation hiatuses, but the possibility
that precipitation is needed to preserve the BFLs after formation. Therefore, weak relationships between BFLs and individual
atmospheric processes at the seasonal scale does not mean that this atmospheric process is not related to BFLs. In addition,
based on the studies of BFLs in other ice cores presented in the introduction (Fujii and Kusunoki, 1982; Albert et al., 2004;
Fitzpatrick et al., 2014; Fegyveresi et al., 2018; Weinhart et al., 2021), these processes acting in combination may be important
drivers of BFL formation. On this basis, we conclude that temperature inversion, high precipitation and wind scour can not be
ruled out as contributing to the formation of BFLs at DSS1617. Future research should consider the combined effects of these
four atmospheric processes on BFL formation.

Future work will utilize the SNOWPACK model to further analyse the formation mechanism of BFLs. SNOWPACK is
a vertical, one-dimensional multilayer land-surface snow model, with up to several hundred layers. Based on the research
presented here, we believe that BFLs could form within the upper mm to cm of the snow surface. The large number of model
layers allows accurate simulations at the small surface depths we will target. Based on meteorological data, SNOWPACK
can simulate the development of the snowpack and produce snow profiles. As well as the primary state variables which are
common in general snow models (temperature, density, etc), SNOWPACK can also describe snow grain characteristics using
four microstructure parameters, including dendricity, sphericity, grain size and bond size. Because BFLs are characterised as
ice layers without bubbles, their density should be higher than the surrounding snow layers above and below them. Meanwhile,
the snow grain characteristics near where the BFLs are formed should also be different from surrounding snow. Thus, the snow
density and snow grain characteristics in SNOWPACK generated snow profiles would be used to identify simulated BFLs. This
approach can give us a more accurate time of BFL formation so that we can study the surrounding environmental changes and
possible atmospheric processes occurring during BFL formation.

### 4.4 Annual and seasonal time series of synoptic types

From Table 4, BFL occurrence per year is significantly correlated with the total number of days per year where both SOM1
($r_a$=0.383, $p_a$=0.048) and SOM2 ($r_a$=-0.523, $p_a$=0.005) are the dominant synoptic type, suggesting the BFL record may
represent large scale atmospheric circulation to some degree. SOM1 represents a synoptic pattern where a strong positive
geopotential height anomaly is located at $\sim$ 55°S, 115°E (to the north of Law Dome), with two negative height anomalies
located to the east and west (Pohl et al., 2021; Udy et al., 2021). SOM2 shows a negative geopotential height anomaly at $\sim$
55°S, 90°E (to the northwest of Law Dome), with one positive height anomaly located to the east (Udy et al., 2021). The



cyclones over the southern Indian and southwest Pacific Oceans are the main controlling factors of precipitation variation and extreme precipitation events for coastal East Antarctic regions like Law Dome (Uotila et al., 2011; Catto et al., 2015; Udy et al., 2022). During days with SOM1, the positive geopotential height anomaly blocks the passage of moist maritime air to Law Dome, leading to lower than average daily precipitation. In contrast when a SOM2 pattern predominates with the positive geopotential height anomaly located further east, moist maritime air masses from the mid-latitudes are channeled southward along the eastern boundary of the negative anomaly to the East Antarctic coastline and plateau regions, which favours increased snowfall at Law Dome (Udy et al., 2022). This pattern of higher and lower than average daily precipitation values is likely related to the contrasting relationships we have found with accumulation hiatuses and high precipitation events. While we are unable at this stage to disentangle the two processes, what does appear to be clear is that BFLs at Law Dome are related to meridional water vapour transport in the southern Indian Ocean and appear to record related precipitation events in the Law Dome region.

As the the SOM1 and SOM2 anomalies have been defined over a large region of the southern Indian Ocean south of the subtropics (Udy et al., 2021, 2022), the correlations between BFLs and synoptic types not only indicate that BFLs have the potential to preserve large scale atmospheric circulation, but may also be relevant to climate variability in the SW Pacific and Australia. In previous studies, van Ommen and Morgan (2010) found a significant inverse correlation between the annual precipitation from the Law Dome ice core and observed annual precipitation in southwest Western Australia. In other studies, summertime sea salt aerosol concentrations and annual accumulation in the Law Dome ice core reflect annual rainfall variability in eastern Australia and Pacific decadal variability (Vance et al., 2013, 2022). The synoptic scale mechanism supporting this relationship has recently been defined (Udy et al., 2022), reinforcing the utility of the Law Dome record for lower latitude climate risk analyses (e.g. see Tozer et al. (2018); Armstrong et al. (2020); Kiem et al. (2020)). While these studies demonstrate the connection between the Law Dome ice core record and Australian climate, they focus primarily on annual or seasonal variability in accumulation and trace aerosol concentrations. In contrast, BFLs may be a proxy for single precipitation-related events. Developing a new BFL record, and importantly, defining the mechanisms behind BFL formation and preservation will be important for future studies of past polar synoptic variability as well as southern Indo-Pacific climate variability, with links to lower latitudes. In future research, snow pack models will be used to further elucidate the mechanism/s behind formation of BFLs, which will in turn inform on the use of this record as a proxy for event-scale precipitation at Law Dome, and in other ice core records from Antarctica and Greenland.

## 5  Conclusion

This study aimed to investigate newly discovered physical structures identified as bubble-free layers (BFLs) in the Law Dome DSS1617 ice core, and to explore their association with local weather processes and large-scale atmospheric circulation. From this research, we found that BFLs are thin ice layers without bubbles in the ice core. By comparing with previous studies, we believe that the DSS1617 BFLs are neither a bubble-free cloudy band nor a melt layer, but rather are most likely a crust formed on or near the surface of the snow pack. From the BFL time series spanning 1990-2016, we found that BFL occurrence has



pronounced seasonality, occurring mainly in autumn and winter. This seasonality suggests that BFL formation may be climate-related, and have the potential to be developed as a new climate proxy. The seasonality of DSS1617 BFLs is different from that of other Antarctic ice cores reported in previous studies that have shown BFL peaks in summer and early autumn. This suggests that the formation and/or preservation mechanism of DSS1617 BFLs may be different from BFLs from other locations. By exploring BFL occurrence and local weather processes using estimated intra-seasonal dating and a simple correlation analysis, we found that BFL formation is likely to be related to, or co-occur with, accumulation hiatuses of 2 days or more and up to weeks. We also found that BFLs may be associated with synoptic variability in the southern Indian Ocean, specifically meridional patterns in the mid-latitudes that either block or channel moisture to the ice core site (Udy et al., 2021, 2022). This suggests that BFLs may record atmospheric processes in the Law Dome region and have the potential to shed light on past variability of large-scale atmospheric circulation in the southern Indian Ocean.

A better understanding of BFLs is important for ice core dating and the palaeoclimate record interpretation not only at Law Dome, but for other high resolution ice core sites. Currently, the primary climate proxies developed from high resolution ice core records are annually resolved, with only a few having clear seasonal markers, such as the summer peaks of water stable isotopes and chemical impurities. If BFL records can be developed further back beyond the satellite era, and their mechanism of formation identified, they represent an opportunity to develop a synoptic-scale proxy from the Law Dome ice core spanning recent centuries and even millennia. In this way, BFLs could be among the first proxies to quantify the frequency of individual weather events (e.g. prolonged accumulation hiatuses as identified in this study). Finally, this work suggests that BFLs have the ability to record or represent large-scale atmospheric circulation. BFLs not only can help to improve our understanding of past atmospheric variability in the southern Indian Ocean and the South West Pacific, but may also provide a pathway to the development of new and novel proxies from existing ice core archives.

This study is limited by the simple analysis used to detect the atmospheric processes which may be relevant to BFL formation and/or preservation. The processes needed to form and preserve BFLs are more complex than can be represented in this simple correlation analysis. BFL formation may be a result of a combination of processes, which our analysis can not currently resolve. However, this study is the first attempt to understand the relationship between BFL occurrence in the Law Dome ice core and atmospheric processes. Despite the simple analysis, the results indicate that BFLs have potential value for past climate research.

In future research, we will use layer resolving snow pack models to further investigate the formation and preservation mechanism of BFLs, so as to better understand the mechanisms leading to their occurrence in ice cores. These modelling analyses will help bolster the use and development of BFLs as a new proxy record in the Law Dome ice core, and more broadly. We will also compare BFLs from other ice core records where available to study any commonalities or differences in formation and preservation.

*Data availability.* All data relating to the development of the age-depth scale for the DSS1617 ice cores is available through Jong et al. (2022). ERA5 data was access from Copernicus Climate Change Service (https://cds.climate.copernicus.eu/), last date of access 12/12/2022.





Automatic Weather Station data was accessed from the Australian Antarctic Division (AAD) network (http://aws.cdaso.cloud.edu.au/). The synoptic typing dataset can be accessed through Udy et al. (2021).

**Appendix A**

This appendix contains supplementary figures relevant to the study.

**A1    Reliability of using ERA 5 precipitation to date monthly accumulation at Law Dome**

Here we conduct an additional test for the reliability of ERA5 snowfall data, by comparing ERA5 cumulative precipitation with AWS snow surface height. From Figure A1, it is clear that ERA5 can represent the trend of snow surface increases relatively
well, with most of the rapid increases in surface height in the AWS data represented in the ERA 5 data. While the absolute values are not well represented by ERA 5, it is the relative change that was needed in this study to apportion the cumulative precipitation at the ice core site in order to produce the sub-annual dating.

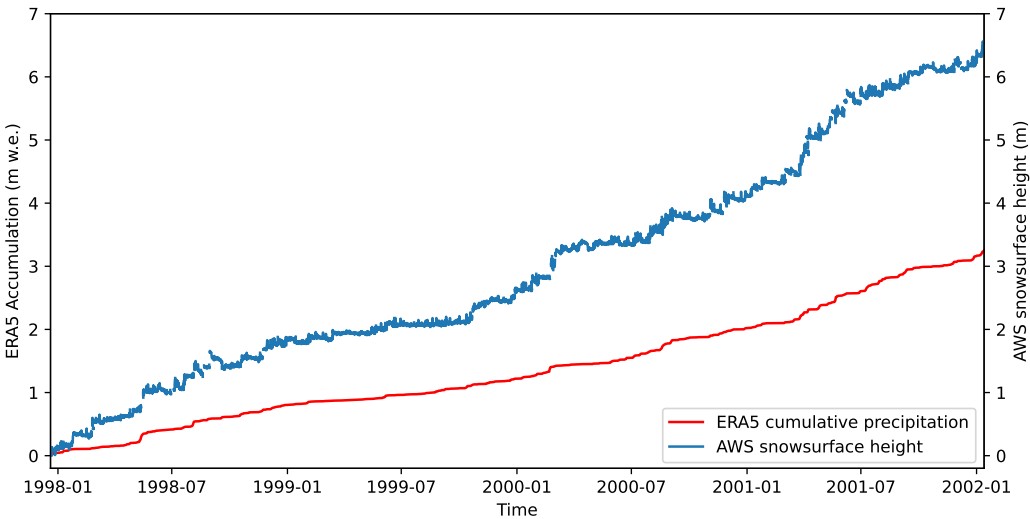

**Figure A1.** Comparison between ERA5 cumulative precipitation (red) and AWS snow surface height time series (blue) from 1998 to 2001.

**A2    Time series of atmospheric processes**

Figure A2 and Figure A3 show the time series of BFL-related atmospheric processes. The correlation coefficient between the
520 time series of BFLs and each atmospheric process is in Table 3.





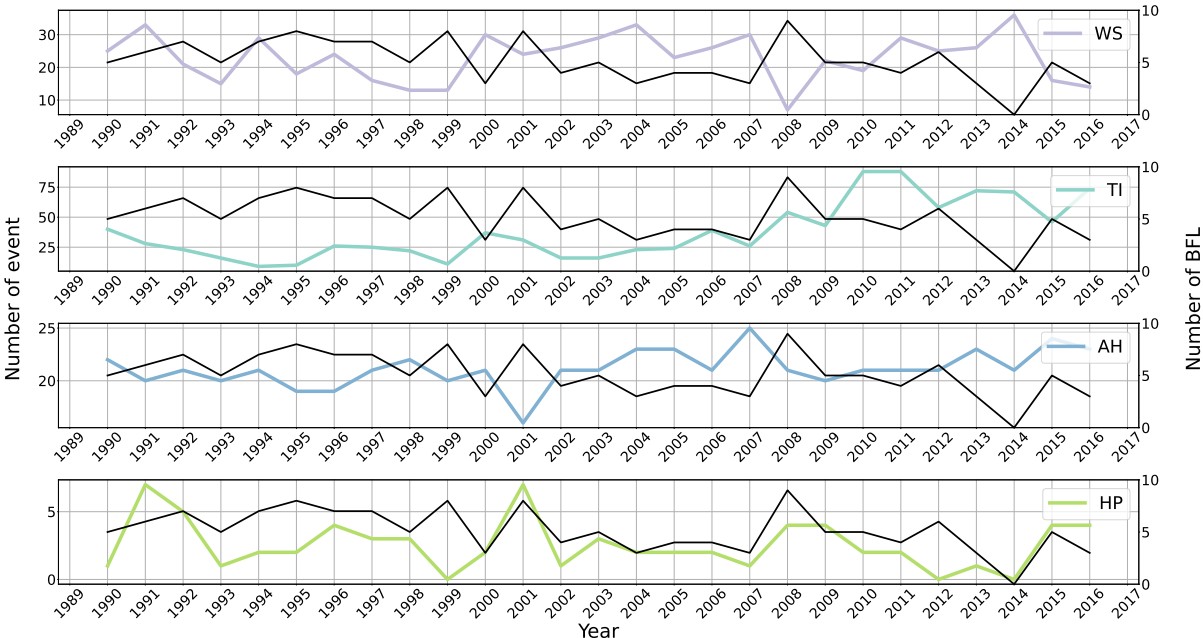

**Figure A2.** Time series of annual number of BFLs (black) and atmospheric events, where WS is wind scour events, TI is temperature inversions, AH is accumulation hiatuses and HP is high precipitation events.

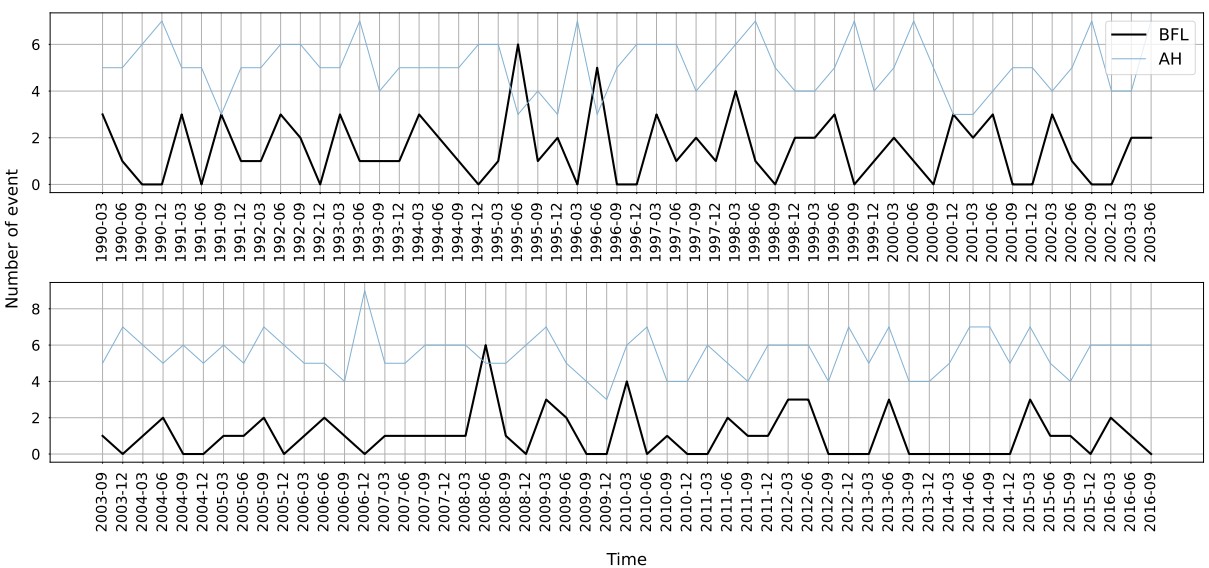

**Figure A3.** Time series of number of BFLs (black) and number of accumulation hiatuses (blue) by season.



## A3 Time series of synoptic types

Figure A4 shows the time series of BFL-related synoptic types. The correlation coefficients between the time series of BFLs and each synoptic type is in Table 4.

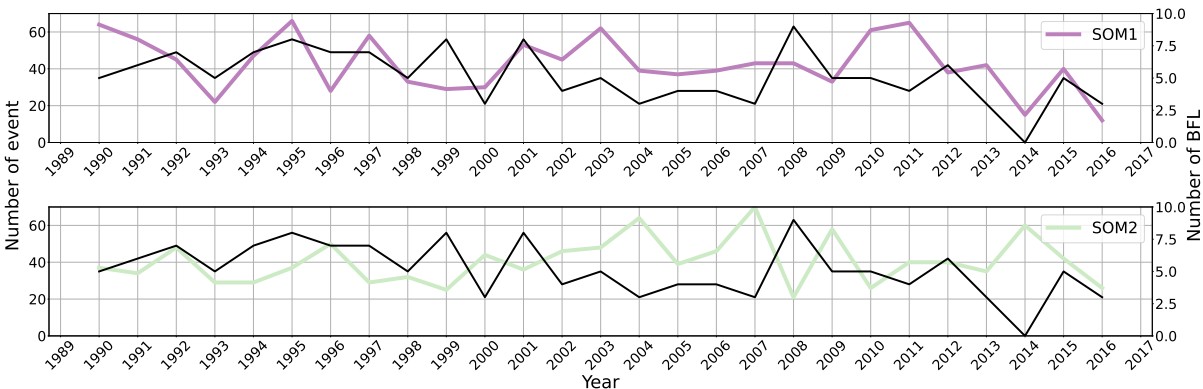

**Figure A4.** Time series of the annual number of BFLs (black) and annual count of the two synoptic types (SOM 1 and SOM 2) described in this study from Udy et al. (2021).

*Author contributions.* LZ designed the analysis and wrote the paper with contributions from all co-authors.

*Competing interests.* The authors declare no competing interests.

*Acknowledgements.* This project received grant funding from the Australian Government as part of the Antarctic Science Collaboration Initiative program. LZ is supported by an Australian Research Training scholarship and an Australian Antarctic Program Partnership top-up scholarship. This work contributes to an Australian Research Council Discovery Project (DP220100606) and Australian Antarctic Science projects 4414, 4537, and 4061.



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
