# Peer review of "Identifying atmospheric processes favouring the formation of bubble free layers in Law Dome ice core, East Antarctica"

_EGUsphere, 2023_

## Referee Comment (RC1)

This paper is very well researched and I am fascinated by the exploration of bubble free layers that has been carefully presented here. I think this paper is very suitable for publication in this issue. I am especially impressed with the diligence put into testing and developing this new proxy and I agree with the authors that it holds potential for future study and may contain valuable climate information. I also appreciate the authors' careful and honest interpretation of their results and the associated limitations. The writing quality of the paper is excellent.

Below I have outlined a list of specific comments that I hope will be useful to the authors in improving and revising the manuscript for final publication. Many of these items are suggestions or minor points. At a broad level, my primary critique is that I think some of the interpretations and discussion should be further developed. While I don't disagree, I don't think that some of the statements linking BFLs to broader climate or to depth of formation are as well-supported by the text and discussion as they could be. I also think that you can leverage the ERA5 dataset and all the work you've done to demonstrate its efficacy here to a much greater effect. The discussion in particular will improve substantially with some further analysis that refines and tests relationship between BFLs, snow accumulation and geopotential height fields using reanalysis data. I also suggest that you capitalize on the fact you have overlapping data from both reanalysis and an AWS on-site. Right now, the paper treats the AWS data and ERA5 data mostly in parallel, but I think some opportunity exists for more integration and validation between these datasets.

I hope the comments below are useful and I am grateful to the authors for developing this new and fascinating dataset. It is my recommendation that this manuscript be accepted for publication with minor revisions.

Sincerely,
Dominic Winski

Figure 1: The latitudes and longitudes in panels A and B on Figure 1 do not match.

Line 73: "Upon physical inspection of the DSS1617 core, BFLs were found to occur in almost all cores" is a little confusing in this context. I suggest "found in almost all 1-meter core segments" or something.

Line 97: Need space before reference

Line 126: Pixels per image? Pixels per cm^2?

Fig. 2 caption: Check with the journal, but I'm guessing you'll need to change the next sentence and provide an actual DOI before the next round. Lots of data repositories can hold a link for you and only go 'live' upon publication or acceptance.

Line 135: Can you explain why higher isostatic pressure would lead to a smoother boundary?

Line 177: It's nice your statistical metrics come out significant, but you only have a sample size of 4 years. I'd be much more convinced if you break this down into all 48 months and show

those correlations. That will also help validate your choice of monthly time horizons and accumulation values. The data in Figure A1, for example, looks like it has a very high correlation at much finer time resolution.

Figure 4: This is commendable and certainly an improvement over using equal-intervals to assign months. However, as you are aware, accumulation patterns especially on month-to-month timescales can vary even over scales of meters. Have you made any effort to assess the accuracy of you monthly picks using data in the ice cores – chemistry with a well established seasonality for example. You can also use the weather station to validate the monthly snow accumulation too. Later on, you back off and end up only using seasons. So it would help to have some quantification of uncertainty regardless so that you have strong justification for whatever resolution you ultimately use to interpret your dataset.

Line 191: I'm glad you are conscious of the limitations here. Similar to my last comment, it would be helpful to quantify this if possible. I think you've got all the information you need to do this given the overlap with the AWS and previously defining seasonality in chemistry.

Lines 212-217: Excellent!

Figure 5: The gray bar isn't visible unless I zoom in to 400%. I suggest either changing the symbology or using a less clustered example.

Line 230-232: I was a little confused by this sentence, since it seems to contradict the previous sentence, until I read on to the next paragraph. I suggest rewording/reorganizing for clarity.

Line 233-239: I can tell you are trying to be very clear and precise in your writing here, but I'm afraid I'm still a little confused. The examples that follow help a little, but I suggest revisiting this section and consider adding an equation, figure or conceptual flowchart or something.

Line 256: TIS or tis? DT1 or dT1?

Lines 258-267: You could use your AWS for this analysis too, right? Do you get similar answers for 1998-2001 with the AWS data vs. the ERA5 data?

Line 291: So you are dating each BFL to within a month (as in Fig. 4), but you are only comparing to other meteorological variables on a seasonal basis, right? Stating earlier that your interpretive goal is only resolved to seasons (4/year) rather than months (12/year), might be helpful and would've allayed some of my earlier worries.

Lines 309-310: I'm not sure that I am convinced of this statement. Because of the translation you are making from depth in the core to accumulated snow from your AWS, you will necessarily intersect with the AWS accumulation curve most often during periods of large accumulation increases. This might be fine if your AWS were at the core site itself, but unless the snowfall here is uncommonly uniform, I would certainly expect cm-scale differences in snow accumulation and redistribution over a few hundred meters. Looking at figure 7, it appears that differences of a few cm in one direction or another might affect your conclusions about the

conditions under which BFLs are most likely to form.  My suggestion would be to do your best to quantify the error associated with your assumptions in section 2.4.  I know this is difficult to estimate, but any estimate would be a start (plus your thinking along these lines so far as been skilled and thoughtful).  Then with these error estimates, I suggest doing some sort of sensitivity study using your approach in Fig. 7.  If your depth/time estimates are off by X%, how many BFLs occur during storms vs. hiatuses for different scenarios within your error range?

Section 3.3/Table 3: Just to check – by seasonal you mean 4 values/year, right?

Line 330: Should be Table 4.

Line 338: Suggest replacing 'numerous' with 'two'.

Line 347: Can you expand on this idea?  Why would summer/autumn BFL formation be climate-related as opposed to winter formation being surface process related?  How does seasonality alone suggest a climate-related process?  Connecting this logical framework is important, because your claim in the following sentence that "BFLs have the potential to provide past climate information" is critical to the paper.

Line 357-375: Much of these paragraphs repeat from earlier.  My suggestion would simply be to mention up front that you will only interpret to the detail level of seasons and lay out your justification then, in the methods.  This will save time in the discussion and show readers earlier that you are aware of your dating limitations early on.

Line 405-424: I suggest condensing/reducing this paragraph significantly.

Line 425-426: Why do you conclude that the BFLs are formed so close to the surface?  This was never really discussed.

Line 449-454: You don't have to hedge here – you have all the data in the ERA5 dataset you need to say something really concrete.  Does a correlation between accumulation at Law Dome and 500 mb GPH produce fields similar to SOM1 or SOM2?  If you think moisture transport is important, ERA5 has variables you can use in your analysis.  You could also isolate times during accumulation hiatuses and determine how atmospheric characteristics during hiatuses differ from baseline circulation patterns.  Finally, if you want to relate BFLs to regional climate patterns, you should at least consider running correlations between BFLs in your ice core and different ERA5 variables.  This will add a lot more specificity and detail to the discussion overall.  Lots of this can be done in 10-15 minutes using online web apps.  I use https://climatereanalyzer.org.

Line 465-469: This feels a little more like a thesis proposal than a paper.  All you have to say is that if the mechanism behind BFL formation can be determined (perhaps through modeling) then they may have potential as a climate proxy.

Conclusions: Since all of this repeats from previous material, I suggest removing, greatly reducing, or bolstering the discussion with additional information.

Figure A2: This figure is very important – at least panel AH, which you conclude is most meaningful.  I suggest putting this into the main body of the paper.  This is potentially true for A3 and A4 as well.  Also, not having the Udy et al. 2021 paper on hand, I wouldn't mind a plot showing the SOM1 and SOM2 patterns, at least in the appendix.

---

## Author Comment (AC1)

This paper is very well researched and I am fascinated by the exploration of bubble free layers that has been carefully presented here. I think this paper is very suitable for publication in this issue. I am especially impressed with the diligence put into testing and developing this new proxy and I agree with the authors that it holds potential for future study and may contain valuable climate information. I also appreciate the authors' careful and honest interpretation of their results and the associated limitations. The writing quality of the paper is excellent.

Below I have outlined a list of specific comments that I hope will be useful to the authors in improving and revising the manuscript for final publication. Many of these items are suggestions or minor points. At a broad level, my primary critique is that I think some of the interpretations and discussion should be further developed. While I don't disagree, I don't think that some of the statements linking BFLs to broader climate or to depth of formation are as well- supported by the text and discussion as they could be. I also think that you can leverage the ERA5 dataset and all the work you've done to demonstrate its efficacy here to a much greater effect. The discussion in particular will improve substantially with some further analysis that refines and tests relationship between BFLs, snow accumulation and geopotential height fields using reanalysis data. I also suggest that you capitalize on the fact you have overlapping data from both reanalysis and an AWS on-site. Right now, the paper treats the AWS data and ERA5 data mostly in parallel, but I think some opportunity exists for more integration and validation between these datasets.

I hope the comments below are useful and I am grateful to the authors for developing this new and fascinating dataset. It is my recommendation that this manuscript be accepted for publication with minor revisions.

Sincerely, Dominic Winski

We thank Dr Winski for the encouraging and constructive comments.

Figure 1: The latitudes and longitudes in panels A and B on Figure 1 do not match.

We will replot this figure.

Line 73: "Upon physical inspection of the DSS1617 core, BFLs were found to occur in almost all cores" is a little confusing in this context. I suggest "found in almost all 1-meter core segments" or something.

We agree. We will rewrite this sentence as suggested.

Line 97: Need space before reference

We will add a space there.

Line 126: Pixels per image? Pixels per cm^2?

We will clarify how the resolution of the image is determined, as this statement as it currently reads is misleading. Instead of:

'Ice samples were imaged by a computer-controlled line scan camera at a resolution of 2,048 pixels.'

This sentence should read:

'The images produced by the ILCS have a resolution of 19,932 x 2,048 pixels.'

We will change this in the revised version.

Fig. 2 caption: Check with the journal, but I'm guessing you'll need to change the next sentence and provide an actual DOI before the next round. Lots of data repositories can hold a link for you and only go 'live' upon publication or acceptance.

We are finalizing the data repository process, and will provide a DOI in the revised manuscript where the datasets produced in this study will be freely available.

Line 135: Can you explain why higher isostatic pressure would lead to a smoother boundary?

In the deep core, the high isostatic pressure will compress the BFLs, forming a denser bubble free ice layer with smoother boundaries. We will add this detail to the section at line 135.

Line 177: It's nice your statistical metrics come out significant, but you only have a sample size of 4 years. I'd be much more convinced if you break this down into all 48 months and show those correlations. That will also help validate your choice of monthly time horizons and accumulation values. The data in Figure A1, for example, looks like it has a very high correlation at much finer time resolution.

We agree. We will make four plots to show the monthly accumulation in each year in 1998-2001. The correlation coefficients between ERA5 and AWS accumulation for all 48 months will also be calculated for the revised manuscript.

Figure 4: This is commendable and certainly an improvement over using equal-intervals to assign months. However, as you are aware, accumulation patterns especially on month-to-month timescales can vary even over scales of meters. Have you made any effort to assess the accuracy of you monthly picks using data in the ice cores – chemistry with a well established seasonality for example. You can also use the weather station to validate the monthly snow accumulation too. Later on, you back off and end up only using seasons. So it would help to have some quantification of uncertainty regardless so that you have strong justification for whatever resolution you ultimately use to interpret your dataset.

Unfortunately, we only have the 4 years shown of AWS snowfall data, so we can't validate the monthly accumulation with the AWS over the full dataset. Because of the limitation, we opted to only use the more conservative approach of seasonal data.

While Law Dome has very high accumulation, we still don't think we can accurately predict monthly changes from the chemistry or isotope data. We do have some measures of sub-annual dating at Law Dome, especially in sea salts, however we

would advise caution in over-interpreting seasonal dating, and we don't advocate monthly dating using chemistry. This further reinforces our decision to stick with seasonal dating. We acknowledge that we weren't totally clear about our use of monthly versus seasonal data in the paper, and we plan to clarify our writing to rectify this.

Line 191: I'm glad you are conscious of the limitations here. Similar to my last comment, it would be helpful to quantify this if possible. I think you've got all the information you need to do this given the overlap with the AWS and previously defining seasonality in chemistry.

On re-reading this section, we think the section is quite confusing - our apologies to the readers/reviewers! We we will change the section from lines 189-192 to read:

'Then, based on the estimated monthly dates transposed to the ice core accumulation series, approximate depths aligned to the commencement of each month were produced by adding up the scaled monthly accumulation amounts sequentially from the beginning of each year (Fig.4). We have high confidence in the annual (summer) layer dating in the DSS1617 ice core data (Jong et al., 2022), due to clear chemical and isotopic signatures. However, we have less confidence in being able to define 'within year' dates using chemistry or isotopes (especially at the monthly scale) thus we have used the ERA 5 accumulation scaled to Law Dome annual accumulation to date within year depths.'

As with the previous comment, we are unable to quantify the error in our monthly dating using chemistry or isotopic analyses from the ice core, because we only have complete confidence in their ability to depict annual layers, or in some rare circumstances seasonal features (e.g. low sea salt concentrations during summer.

Lines 212-217: Excellent!

Thank you!

Figure 5: The gray bar isn't visible unless I zoom in to 400%. I suggest either changing the symbology or using a less clustered example.

We will change the gray bar into more prominent colors.

Line 230-232: I was a little confused by this sentence, since it seems to contradict the previous sentence, until I read on to the next paragraph. I suggest rewording/reorganizing for clarity.

We agree that this is confusing and will reword to clarify that we are talking about two different scales of processes - the southern Indian ocean, and processes that are occurring over 10's to 1000's of metres at Law Dome itself.

Line 233-239: I can tell you are trying to be very clear and precise in your writing here, but I'm afraid I'm still a little confused. The examples that follow help a little, but I suggest revisiting this section and consider adding an equation, figure or conceptual flowchart or something.

We will add a flowchart here to clarify.

'TIS' is temperature inversion strength. We calculated TIS for every 6h timestep. 'tis' is the threshold we calculated, whereby a temperature inversion event is identified by TIS > tis. We will revisit this section to clarify.

Lines 258-267: You could use your AWS for this analysis too, right? Do you get similar answers for 1998-2001 with the AWS data vs. the ERA5 data?

The AWS data is only available for four years, compared to the 30 years of ERA 5 over the ice core period. We do compare the AWS to ERA 5 over the AWS timescale (Fig A1) and they match very well. However, we don't think we should use the AWS data for this analysis. This is because the AWS data contains a lot of missing data (and this may be biased due to the snow surface height sensor being overloaded during high precipitation events). In addition, the snow surface height data varies in both directions due to settling, ablation, redistribution etc etc. Thus, it would not be straightforward to interpret this data as suggested. The ERA 5 accumulation data is longer, has no missing data, and has the temporal resolution that we need. Ultimately, while the AWS data seems like the best quantitative dataset for the purpose, we think it is best in this study to treat it as qualitative data.

Line 291: So you are dating each BFL to within a month (as in Fig. 4), but you are only comparing to other meteorological variables on a seasonal basis, right? Stating earlier that your interpretive goal is only resolved to seasons (4/year) rather than months (12/year), might be helpful and would've allayed some of my earlier worries.

We will rewrite this part to make it more explicit as per our comments previously. We agree we weren't clear about why we opted to interpret only at the seasonal level, so we will revisit the methodological sections to ensure we are clear about our seasonal investigations. We will also revisit the earlier part of this paragraph to clarify.

Lines 309-310: I'm not sure that I am convinced of this statement. Because of the translation you are making from depth in the core to accumulated snow from your AWS, you will necessarily intersect with the AWS accumulation curve most often during periods of large accumulation increases. This might be fine if your AWS were at the core site itself, but unless the snowfall here is uncommonly uniform, I would certainly expect cm-scale differences in snow accumulation and redistribution over a few hundred meters. Looking at figure 7, it appears that differences of a few cm in one direction or another might affect your conclusions about the conditions under which BFLs are most likely to form. My suggestion would be to do your best to quantify the error associated with your assumptions in section 2.4. I know this is difficult to estimate, but any estimate would be a start (plus your thinking along these lines so far as been skilled and thoughtful). Then with these error estimates, I suggest doing some sort of sensitivity study using your approach in Fig. 7. If your depth/time estimates are off by X%, how many BFLs occur during storms vs. hiatuses for different scenarios within your error range?

We think that this comment really gets to the core of this study, and what the next direction should be. However we disagree with the reviewer about the next steps. While Law Dome is probably an ideal ice core to study BFLs given its high annual accumulation rate and the fact it is a well studied record, even at Law Dome differences of a few centimeters of snowfall will completely affect the *apparent* date of BFL occurrence. While we could try our best to quantify the uncertainties, we have opted to provide qualitative uncertainty by using the red and grey bands in Fig. 7 which give our qualitative estimate of the dating uncertainty. We don't think the data we currently have at hand would allow further refining of error around dates in a robust way.

Ultimately, this study is an exploratory investigation of events that coincide with BFL occurrence, and we make pains to differentiate between BFL *occurrence* compared to *formation*. A mechanistic study is required to investigate the *formation* of BFLs (and their preservation). We maintain that the next best step to investigate the mechanism of formation (and preservation) is a modelling study using the SNOWPACK model, within which we can simulate densification, vapour transport, etc. At this point, we will be able to re-visit the datasets from this paper with a clearer understanding of how to estimate uncertainty, because we will have an understanding of the atmospheric and snowpack processes that must be present to form BFLs. This will allow us to better define the climate process/es that BFLs represent.

Section 3.3/Table 3: Just to check – by seasonal you mean 4 values/year, right?

Yes. We will clarify this in Table 3 caption.

Line 330: Should be Table 4.

Yes, We will change this.

Line 338: Suggest replacing 'numerous' with 'two'.

Yes, we will change as suggested.

Line 347: Can you expand on this idea? Why would summer/autumn BFL formation be climate- related as opposed to winter formation being surface process related? How does seasonality alone suggest a climate-related process? Connecting this logical framework is important, because your claim in the following sentence that "BFLs have the potential to provide past climate information" is critical to the paper.

We are a bit unclear about the reviewers comment here, as Law Dome BFLs occur in autumn/winter, and we are saying that Law Dome (autumn/winter) BFLs are climate related (not surface process related). Summer BFLs (e.g. WAIS) may be surface process related, if they are local radiation/melt layers. This is harder to understand at Law Dome, where BFLs predominantly occur in autumn/winter (unlikely to have a strong radiation/melt link).

On re-reading this paragraph however, we think our wording was a little vague, and gave an unintended meaning to our statements. In particular, this sentence:

"This seasonality suggests that BFLs may be related to regional climate variability, rather than local surface processes."

We propose changing this sentence to:

"This seasonality suggests that BFLs may be related to *seasonal* climate variability, rather than *localised* surface processes."

Line 357-375: Much of these paragraphs repeat from earlier. My suggestion would simply be to mention up front that you will only interpret to the detail level of seasons and lay out your justification then, in the methods. This will save time in the discussion and show readers earlier that you are aware of your dating limitations early on.

We will add line 357-372 to the methods to clarify the ice core monthly dating limitation. In line 373-375, we would like to point out that DSS1617 BFLs seasonality is different from BFLs in previous studies, and this difference does not come from  different dating strategies. So we will keep line 373-375 here. We will however, review this entire section for clarity and brevity.

Line 405-424: I suggest condensing/reducing this paragraph significantly.

We will reduce this part.

Line 425-426: Why do you conclude that the BFLs are formed so close to the surface? This was never really discussed.

Previous crust studies have suggested that those crusts should form at, or below the snow surface.  In this study, the hypothesis is that the DSS1617 BFLs are formed through the vapor condensed at the cold snow surface. Furthermore, scientists who have worked around the DSS sites have qualitatively observed there are thin ice crusts at the snow surface. These ice crusts could be newly formed BFLs.  Thus, we suggest the BFLs may be formed close to the snow surface.

We will clarify this in the discussion.

Line 449-454: You don't have to hedge here – you have all the data in the ERA5 dataset you need to say something really concrete. Does a correlation between accumulation at Law Dome and 500 mb GPH produce fields similar to SOM1 or SOM2? If you think moisture transport is important, ERA5 has variables you can use in your analysis. You could also isolate times during accumulation hiatuses and determine how atmospheric characteristics during hiatuses differ from baseline circulation patterns. Finally, if you want to relate BFLs to regional climate patterns, you should at least consider running correlations between BFLs in your ice core and different ERA5 variables. This will add a lot more specificity and detail to the discussion overall. Lots of this can be done in 10-15 minutes using online web apps. I use https://climatereanalyzer.org.

We are encouraged by the faith in our study! However, given our response to comments above about lines 309-310, we think we still do have to hedge a bit here, and we think the next best step is to switch to the mechanistic investigation of formation using the snowpack model.

In addition, an in depth study of accumulation at the Law Dome and Mount Brown South ice core sites during different synoptic types/situations is currently well-advanced and will be submitted elsewhere (Udy et al., in prep). This work will be very complementary to this study, and will certainly aid in the development of the BFL proxy, so we would prefer to develop this work in depth as a standalone manuscript, rather than add it as a small section here.

Line 465-469: This feels a little more like a thesis proposal than a paper. All you have to say is that if the mechanism behind BFL formation can be determined (perhaps through modeling) then they may have potential as a climate proxy.

We will reduce/clarify this section, however we would prefer to keep some of the discussion of the snowpack model, as we are actively working on using SNOWPACK to investigate BFL formation.

Conclusions: Since all of this repeats from previous material, I suggest removing, greatly reducing, or bolstering the discussion with additional information.

We agree. We will reduce/clarify this section.

Figure A2: This figure is very important – at least panel AH, which you conclude is most meaningful. I suggest putting this into the main body of the paper. This is potentially true for A3 and A4 as well. Also, not having the Udy et al. 2021 paper on hand, I wouldn't mind a plot showing the SOM1 and SOM2 patterns, at least in the appendix.

We will move figure A2, A3 and A4 in the main body. We will also add a schematic plot describing SOM1 and SOM2.

---

## Author Comment (AC3)

This paper takes a very detailed look at bubble free layers in the Law Dome ice cores and their potential drivers. A particular care is given to the identification of the timing of the occurrence of these layers, with respect to the season, which is essential for the correlation with climate variables like temperature, accumulation (or hiatus), and atmospheric circulation.

I am impressed with the level of details and care given in the paper. Not much more could have been done.

In the reading, I still struggle a bit with how things are presented, and in the section describing the exploration of mechanisms, some extra care could be put on describing hypotheses more clearly, and then, what data can be brought to support them or not.

A diagram/drawing showing the potential formation mechanisms would greatly help the reading of the paper.

We do appreciate that this would add to the reading of the paper but it is not something that we think we can add to this manuscript. As described in the introduction, this is a descriptive and exploratory study. The next stage of our work is to use the SNOWPACK model to investigate possible formation mechanisms using the analysis we present in this paper as a starting point. The data we present and analyze here is occurrence of BFLs rather than formation. Currently, we don't know how frequently the mechanisms to produce BFLs occur without the preservation of a BFL (as preservation could be decoupled at times from formation, e.g. due to ablation of a newly formed layer). Equally, there could be a timelag between formation and apparent preservation. We will clarify the text throughout the manuscript to ensure we differentiate clearly between BFL occurrence (studied in this work) and BFL formation/preservation (which we are currently commencing with the snowpack model, and intend to publish as a mechanistic study).

These are mostly cosmetic comments, on what is an excellent, thorough paper. I suggest minor revisions.

We thank the reviewer for their helpful and constructive comments.

line 35: bubble number density and paleoclimate: cite Fegyveresi 2016: https://agupubs.onlinelibrary.wiley.com/doi/full/10.1002/2015PA002851, also potentially:

Spencer, M. K., R. B. Alley, and J. J. Fitzpatrick (2006), Developing a bubble number-density paleoclimatic indicator for glacier ice, J. Glaciol., 52(178), 358– 364, doi:10.3189/172756506781828638.

Fegyveresi, J. M., R. B. Alley, M. K. Spencer, J. J. Fitzpatrick, E. J. Steig, J. W. C. White, J. R. McConnell, and K. C. Taylor (2011), Late-Holocene climate evolution at

the WAIS Divide site, West Antarctica: Bubble number-density estimates, J. Glaciol., 57(204), 629– 638, doi:10.3189/002214311797409677.

Melt layers: consider citing Keegan 2014: www.the-cryosphere.net/8/1801/2014/ doi:10.5194/tc-8-1801-2014

We will add these references in the paper in the appropriate sections.

line 97: missing space before citation.

We will add space there.

line 225: When you introduce the synoptic types, in section 2.5, you can include here the description of the two important types that you will use later, with a figure (potentially in the supplement if you feel you have too many figures) of the synoptic types, and their relationship with accumulation.

We will add the description to the introduction and a schematic plot of SOM1 and SOM2 in section 2.5.

In general, I think that a description of the relationship between synoptic type and accumulation (or hiatus) is missing in a quantitative sense.
As I understand we could make a schematic:
 synoptic type --causes --> accumulation or hiatus --causes--> bubble free layer.  In this causal chain, you would find a causal relationship between synoptic type and BFL, but you would not be learning anything more about their formation than when you were looking at accumulation/hiatus.

We tried to detect the relationship between synoptic type and accumulation (or hiatus) quantitatively,  but the results were not as definitive as we had hoped. We think this is because we cannot (until we have done the mechanistic modelling currently underway with SNOWPACK) quantitatively define the formation date for BFLs. That is - we cannot with current data distinguish between the dates of formation and preservation. With our current investigations using the SNOWPACK model, we can investigate the BFL formation mechanism, and then we will be able to relate the conditions needed for formation to larger scale phenomena, such as the synoptic types.

Figure 5: I can't see the grey for layer 2

We will change the gray bar into more prominent colors.

Figure 7: the aws data is too tiny, Can you make it a bit thicker? parenthesis issue in caption

We will make the AWS line thicker and rectify the parenthesis issue.

section 3.4

line 330 : you mean table 4

Yes, we will make this correction.

worth adding that som1 is associated with high precip, or describing the significant modes a bit better

We agree, and will add the schematic plot to show SOM1 and SOM2 that we can refer to here.

line 340+:trend: once you have analyzed the drivers, can you say something about the trends in the drivers themselves? Would we be expecting any trend?

We will revisit this section for clarity. However, this section is only describing an apparent (possibly spurious) trend in BFLs through time. We suggest this decreasing trend is an artificially significant trend, possibly due to the surface cores being more fragile and the BFLs in the surface core being more easily broken (and thus missed in the image analysis), rather than the formation of BFLs have a decreasing trend from 1990 to 2016. That is, it is possible that more fragile firn cores closer to the surface break preferentially at BFLs. We are currently analysing longer timeseries of BFL's from other ice core sites, and will be more comfortable discussing any trends once we have the longer timeseries measured and analysed. As suggested, we can then revisit the drivers/mechanisms to see whether they also have trends, as suggested.

line 390: you have here a great comment about formation vs preservation of BFL.

I suggest that you restructure section 4 with the investigations of mechanisms, spelling them one by one, and detailing the supporting data/correlation, and the complicating factors: For instance, make a section on vapor flux upward condensing on the surface, as a formation mechanism, then show evidence for formation during snow-warmer-than-air times, during times of long surface exposure (your hiatus), seasonality, radiation, etc. And discuss evidence where it doesn't quite work.

Repeat this effort for other mechanisms of formation and preservation, aided by schematics, so that we can more clearly see where your thinking is.

We used this structure in a previous draft of this manuscript. After many re-writes, we decided that the structure suggested above was not the best way to structure the discussion. We think this is because we don't know the formation/preservation mechanism yet. We agree that the above suggestion is a nice way to think about the occurrence/formation/preservation of BFLs, however we found that it did not ultimately make for a linear and readable structure in the manuscript sense, largely because this is an exploratory study, and it is difficult to produce schematics of formation, when we have not yet explored formation in the model.

line 419 to 435: not sure it belongs in the paper. start of 4.4 to line 450: should go into the result section.

This section relates a lot to many of the comments from both reviewers requesting more quantitative analysis regarding BFL formation. As we have said, we think we need the model analysis first to do this. As this is the first paper of Lingwei Zhang's

PhD project, we think it is worthwhile keeping this section, as it demonstrates clearly the work we are doing next. However, we will revisit this section to ensure relevance and brevity.

We will revisit the first section of 4.4 to ensure any results are moved to the appropriate section.

Section 4.4: Give more info on the relationship between SOMs and accumulation/hiatus, temperature inversions, wind scours, etc..

We will give more information on the relationship between SOM1, SOM2 and precipitation/hiatuses, however the difference in spatial scales between the synoptic types and the smaller scale processes of wind scours and temperature inversions has not yet been explored (here or elsewhere).

Appendix A1: ERA is overestimating snowfall by a factor of 2. Did you consider scaling ERA, and plotting on figure A1 the scaled ERA to highlight the temporal accuracy of hiatuses, or perhaps even plot the derivative.

in A1, ERA5 is the net accumulation (in metres water equivalent) and the AWS data is the measured snow surface height (including ablation, redistribution, settling etc). They are not really the same thing, so while scaling initially makes sense, it is ultimately misleading (we tried this previously, and felt it was more, rather than less confusing). However we will adjust the axes so that the two datasets can be more easily compared in a visual sense.

---

## Author Response (AR1)

Note - all page and line number references below are for the updated/revised pdf document provided. In addition, the manuscript has been revised elsewhere for minor textual changes to improve english / readability.

**Reviewer 1**

This paper is very well researched and I am fascinated by the exploration of bubble free layers that has been carefully presented here. I think this paper is very suitable for publication in this issue. I am especially impressed with the diligence put into testing and developing this new proxy and I agree with the authors that it holds potential for future study and may contain valuable climate information. I also appreciate the authors' careful and honest interpretation of their results and the associated limitations. The writing quality of the paper is excellent.

Below I have outlined a list of specific comments that I hope will be useful to the authors in improving and revising the manuscript for final publication. Many of these items are suggestions or minor points. At a broad level, my primary critique is that I think some of the interpretations and discussion should be further developed. While I don't disagree, I don't think that some of the statements linking BFLs to broader climate or to depth of formation are as well- supported by the text and discussion as they could be. I also think that you can leverage the ERA5 dataset and all the work you've done to demonstrate its efficacy here to a much greater effect. The discussion in particular will improve substantially with some further analysis that refines and tests relationship between BFLs, snow accumulation and geopotential height fields using reanalysis data. I also suggest that you capitalize on the fact you have overlapping data from both reanalysis and an AWS on-site. Right now, the paper treats the AWS data and ERA5 data mostly in parallel, but I think some opportunity exists for more integration and validation between these datasets.

I hope the comments below are useful and I am grateful to the authors for developing this new and fascinating dataset. It is my recommendation that this manuscript be accepted for publication with minor revisions.

Sincerely, Dominic Winski

We thank Dr Winski for the encouraging and constructive comments.

Figure 1: The latitudes and longitudes in panels A and B on Figure 1 do not match.

We have replotted this figure. The latitudes and longitudes were actually correct, however the angles of the longitudinal lines were visually confusing.

To resolve this, we removed the latitude and longitude from the upper figure, and we provide these details in the text and in the lower figure. See Figure 1, page 4.

Line 73: "Upon physical inspection of the DSS1617 core, BFLs were found to occur in almost all cores" is a little confusing in this context. I suggest "found in almost all 1-meter core segments" or something.

We have rewritten this sentence as suggested. 'Upon physical inspection of the DSS1617 core, BFLs were found to occur in almost all 1-meter core segments.'

See line 74-75 in the revised manuscript.

Line 97: Need space before reference

We added a space there (before (Udy et al., 2021)).

See line 99 in the revised manuscript.

Line 126: Pixels per image? Pixels per cm^2?

We have clarified how the resolution of the image is determined. Instead of:

'Ice samples were imaged by a computer-controlled line scan camera at a resolution of 2,048 pixels.'

This sentence has now become:

'The images produced by the ILCS have a cross-core resolution of approximately 13.6 pixels per mm, and an along-core resolution of approximately 19.8 pixels per mm.'

See line 128-129 in the revised manuscript.

Fig. 2 caption: Check with the journal, but I'm guessing you'll need to change the next sentence and provide an actual DOI before the next round. Lots of data repositories can hold a link for you and only go 'live' upon publication or acceptance.

We have provided the DOI in the revised manuscript where the datasets produced in this study are now archived and freely available.

See line 485 in the revised manuscript.

Line 135: Can you explain why higher isostatic pressure would lead to a smoother boundary?

We added an explanation to clarify this.

'Due to limited isostatic pressure near the surface of the ice sheet, the boundary of the DSS1617 BFLs may not be as smooth as BFLs observed in deep ice.'

See line 139-140 in the revised manuscript.

Line 177: It's nice your statistical metrics come out significant, but you only have a sample size of 4 years. I'd be much more convinced if you break this down into all 48 months and show those correlations. That will also help validate your choice of monthly time horizons and accumulation values. The data in Figure A1, for example, looks like it has a very high correlation at much finer time resolution.

We are now providing a four plots figure (Figure 4) in the revised manuscript to show the monthly accumulation in each year in 1998-2001. The correlation coefficients between ERA5 and AWS accumulation for all 48 months have also been calculated for the revised manuscript (see in Figure 4 caption, line 4).

See Figure 4, page 9 in the revised manuscript.

Figure 4: This is commendable and certainly an improvement over using equal-intervals to assign months. However, as you are aware, accumulation patterns especially on month-to-month timescales can vary even over scales of meters. Have you made any effort to assess the accuracy of you monthly picks using data in the ice cores – chemistry with a well established seasonality for example. You can also use the weather station to validate the monthly snow accumulation too. Later on, you back off and end up only using seasons. So it would help to have some quantification of uncertainty regardless so that you have strong justification for whatever resolution you ultimately use to interpret your dataset.

Unfortunately, we only have the 4 years shown of AWS snowfall data, so we can't validate the monthly accumulation with the AWS over the full dataset. Because of that limitation, we opted to only use the more conservative approach of seasonal data.

While Law Dome has very high accumulation, we still don't think we can accurately predict monthly changes from the chemistry or isotope data. We do have some measures of sub-annual dating at Law Dome, especially in sea salts, however we would advise caution in over-interpreting seasonal dating, and we don't advocate monthly dating using chemistry. This further reinforces our decision to stick with seasonal dating. We acknowledge that we weren't totally clear about our use of monthly versus seasonal data in the paper, and we have clarified our writing. Instead of:

'Although we can produce monthly time series for both BFLs and atmospheric processes, we only produce the correlation between time series at annual and seasonal time scales, due to relatively limited confidence in monthly dating. '

This sentence has now become:

'Although we can produce monthly time series for both BFLs and atmospheric processes, we elected to only report correlations between time series at annual and seasonal time scales, due to lower confidence in monthly ice core dating.'

See line 290-291 in the revised manuscript.

Line 191: I'm glad you are conscious of the limitations here. Similar to my last comment, it would be helpful to quantify this if possible. I think you've got all the information you need to do this given the overlap with the AWS and previously defining seasonality in chemistry.

On re-reading this section, we think the section is quite confusing - our apologies to the readers/reviewers!

Instead of:

'Then, based on this monthly ice core accumulation, approximate monthly boundaries were produced by adding up the monthly accumulations sequentially from the beginning of each year (Fig.4). Due to inherent error in this method (primarily that ERA-5 accurately represents accumulation at the DSS site on monthly scales), the confidence in monthly dating is lower than for the annual dating derived from analysis of seasonally varying chemical and isotopic species.'

We  changed the section from lines 189-192 (in the original manuscript) to read:

'Then, based on the estimated monthly dates transposed to the ice core accumulation series, approximate depths aligned to the commencement of each month were produced by adding up the scaled monthly accumulation amounts sequentially from the beginning of each year (Figure 5). We have high confidence in

the annual (summer) layer dating in the DSS1617 ice core data (Jong et al., 2022), due to clear chemical and isotopic signatures. However, we have less confidence in being able to define 'within year' dates using chemistry or isotopes (especially at the monthly scale) thus we have used the ERA5 accumulation scaled to Law Dome annual accumulation to date within-year depths.'

As with the previous comment, we are unable to quantify the error in our monthly dating using chemistry or isotopic analyses from the ice core, because we only have complete confidence in their ability to depict annual layers, or in some rare circumstances seasonal features (e.g. low sea salt concentrations during summer.)

See line 192-198 in the revised manuscript.

Lines 212-217: Excellent!

Thank you!

Figure 5: The gray bar isn't visible unless I zoom in to 400%. I suggest either changing the symbology or using a less clustered example.

We changed the grey bar into darker grey. We also added an arrow to point out the position of the grey bar, which is narrow, as we wanted to demonstrate the diversity of BFL cases we see.

See Figure 6, page 12.

Line 230-232: I was a little confused by this sentence, since it seems to contradict the previous sentence, until I read on to the next paragraph. I suggest rewording/reorganizing for clarity.

We agree that this is confusing and reworded to clarify that we are talking about two different scales of processes. Then sentence was changed to:

'In addition to large-scale atmospheric circulation, local atmospheric processes that are occurring over 10's to 1000's of metres at Law Dome may also be related to BFL occurrence, including surface-based temperature inversions, wind scour, accumulation hiatuses and high precipitation events.'

See line 235-237 in the revised manuscript.

Line 233-239: I can tell you are trying to be very clear and precise in your writing here, but I'm afraid I'm still a little confused. The examples that follow help a little, but

I suggest revisiting this section and consider adding an equation, figure or conceptual flowchart or something.

We have added a flowchart (Figure 7) here to clarify how we approached the atmospheric process analyses. We also explain this method in the caption of Figure7:

'Thresholds of atmospheric processes were chosen based on the highest correlation coefficients detected in the initial analysis. Then, an objective assessment of whether the threshold was realistic was made, at which point the time series of the specific atmospheric process was generated or, the initial step was redone using the next highest correlation coefficient. This ensured that realistic thresholds that maximised any probable statistical relationships were investigated.'

Figure discussion at line 241 and Figure 7, page 13 in the revised manuscript.

Line 256: TIS or tis? DT1 or dT1?

We have tried to clarify the explanation of our initial and then final threshold selections. 'TIS' is the temperature inversion strength (line 251). We calculated TIS for every 6h timestep. Once we had selected the most appropriate TIS value we termed this our 'tis', which is the final threshold value we used for analysis of the correlation between inversions and BFLs. In the new pdf, we have now written (line 262):
'Among them, the combination of thresholds based on the annual temperature inversion time series most related to BFLs was chosen as the final threshold (tis and dT I , a temperature inversion event is identified by TIS > tis, DT I > dT I ) (see Table 3 for final thresholds).' We have clarified the value of the final threshold (tis, dTI) in Table 3 (column 'Thresholds').

See line 251, line 262, and Table 3 in the revised manuscript.

Lines 258-267: You could use your AWS for this analysis too, right? Do you get similar answers for 1998-2001 with the AWS data vs. the ERA5 data?

The AWS data is only available for four years, compared to the 30 years of ERA 5 over the ice core period. We do compare the AWS to ERA 5 over the AWS timescale (Figure 4) and they match very well. However, we don't think we should use the AWS data for this analysis. This is because the AWS data contains a lot of missing data (and this may be biased due to the snow surface height sensor being overloaded during high precipitation events). In addition, the snow surface height data varies in

both directions due to settling, ablation, redistribution etc etc. Thus, it would not be straightforward to interpret this data as suggested. The ERA 5 accumulation data is longer, has no missing data, and has the temporal resolution that we need. Ultimately, while the AWS data seems like the best quantitative dataset for the purpose, we think it is best in this study to treat it as qualitative data.

Line 291: So you are dating each BFL to within a month (as in Fig. 4), but you are only comparing to other meteorological variables on a seasonal basis, right? Stating earlier that your interpretive goal is only resolved to seasons (4/year) rather than months (12/year), might be helpful and would've allayed some of my earlier worries.

Our goal on BFL dating is to detect the month (12/year) of BFL occurrence. Instead of seasonal and annual scale BFLs dating, in this study, we also present monthly scale dating for BFLs. The number of BFLs by month are shown in Figure 8, panel b. The monthly scale dating for BFLs provided us the occurrence frequency of BFLs in each month.

We produced monthly BFLs time series, but we only reported annual and seasonal correlations between BFLs and atmospheric processes/large scale atmospheric circulations at this early stage of understanding the proxy. We have rewritten this section to make it more explicit that this was our intention.

Instead of:

'Although we can produce monthly time series for both BFLs and atmospheric processes, we only produce the correlation between time series at annual and seasonal time scales, due to relatively limited confidence in monthly dating. '

This sentence has now become:

'Although we can produce monthly time series for both BFLs and atmospheric processes, we elected to only report correlations between time series at annual and seasonal time scales, due to lower confidence in monthly ice core dating.'

See line 290-299 in the revised manuscript.

Lines 309-310: I'm not sure that I am convinced of this statement. Because of the translation you are making from depth in the core to accumulated snow from your AWS, you will necessarily intersect with the AWS accumulation curve most often during periods of large accumulation increases. This might be fine if your AWS were at the core site itself, but unless the snowfall here is uncommonly uniform, I would certainly expect cm-scale differences in snow accumulation and redistribution over a

few hundred meters. Looking at figure 7, it appears that differences of a few cm in one direction or another might affect your conclusions about the conditions under which BFLs are most likely to form. My suggestion would be to do your best to quantify the error associated with your assumptions in section 2.4. I know this is difficult to estimate, but any estimate would be a start (plus your thinking along these lines so far as been skilled and thoughtful). Then with these error estimates, I suggest doing some sort of sensitivity study using your approach in Fig. 7. If your depth/time estimates are off by X%, how many BFLs occur during storms vs. hiatuses for different scenarios within your error range?

We think that this comment really gets to the core of this study, and what the next direction should be. However we disagree with the reviewer about the next steps. While Law Dome is probably an ideal ice core to study BFLs given its high annual accumulation rate and the fact it is a well studied record, even at Law Dome differences of a few centimeters of snowfall will completely affect the *apparent* date of BFL occurrence. While we could try our best to quantify the uncertainties, we have opted to provide qualitative uncertainty by using the red and grey bands in Fig. 9 which give our qualitative estimate of the dating uncertainty. We don't think the data we currently have at hand would allow further refining of error around dates in a robust way.

Ultimately, this study is an exploratory investigation of events that coincide with BFL occurrence, and we make pains to differentiate between BFL *occurrence* compared to *formation*. A mechanistic study is required to investigate the *formation* of BFLs (and their preservation). We maintain that the next best step to investigate the mechanism of formation (and preservation) is a modelling study using the SNOWPACK model, within which we can simulate densification, vapour transport, etc. At this point, we will be able to re-visit the datasets from this paper with a clearer understanding of how to estimate uncertainty, because we will have an understanding of the atmospheric and snowpack processes that must be present to form BFLs. This will allow us to better define the climate process/es that BFLs represent.

Section 3.3/Table 3: Just to check – by seasonal you mean 4 values/year, right?

Yes. We clarify this in Table 3 caption by adding n (line 4: na=27; line 5: ns=107) for annual and seasonal correlation coefficient.

See line 4-5 in table 3 caption, page 18 in the revised manuscript.

Line 330: Should be Table 4.

Yes, We have changed this.

See line 337 in the revised manuscript.

Yes, we have changed as suggested.

See line 345 in the revised manuscript.

Line 347: Can you expand on this idea? Why would summer/autumn BFL formation be climate- related as opposed to winter formation being surface process related? How does seasonality alone suggest a climate-related process? Connecting this logical framework is important, because your claim in the following sentence that "BFLs have the potential to provide past climate information" is critical to the paper.

We are a bit unclear about the reviewers comment here, as Law Dome BFLs occur in autumn/winter, and we are saying that Law Dome (autumn/winter) BFLs are climate related (not surface process related). Summer BFLs (e.g. at the WAIS site) may be surface process related, if they are local radiation/melt layers. This is harder to understand at Law Dome, where BFLs predominantly occur in autumn/winter (unlikely to have a strong radiation/melt link).

On re-reading this paragraph however, we think our wording was a little vague, and gave an unintended meaning to our statements. In particular, this sentence:

'This seasonality suggests that BFLs may be related to regional climate variability, rather than local surface processes.'

We changed this sentence to:

'This seasonality suggests that BFLs may be related to seasonal climate variability, rather than localised surface processes.'

See line 354 in the revised manuscript.

Line 357-375: Much of these paragraphs repeat from earlier. My suggestion would simply be to mention up front that you will only interpret to the detail level of seasons and lay out your justification then, in the methods. This will save time in the discussion and show readers earlier that you are aware of your dating limitations early on.

Here we would like to point out that DSS1617 BFLs seasonality is different from BFLs in previous studies, and this difference does not come from different dating

strategies. Thus we discussed the seasonality and dating strategies for BFLs in previous studies and BFLs in DSS1617 ice core here to demonstrate that the DSS1617 BFLs may be caused by different processes. So we would prefer to keep this section as is.

See line 351-362 in the revised manuscript.

Line 405-424: I suggest condensing/reducing this paragraph significantly.
In this paragraph, because we try to explain the 'weak relationships between BFLs and individual atmospheric processes at the seasonal scale', we repeat our assumptions from time to time. We have now removed a lot of the repeated parts and point out it is a complex process and more research is needed in the future studies.

We have reduced this part into the shorter version.

See line 402-422 in the revised manuscript.

Line 425-426: Why do you conclude that the BFLs are formed so close to the surface? This was never really discussed.

Previous studies have suggested that crusts should form at, or just below the snow surface. Thus, we suggest the BFLs in the DSS1617 site may be formed close to the snow surface. We mentioned the previous crust studies as our evidence in line 424. The description of previous studies can be found at lines 48-64, with line 52 specifically referring to the surface/subsurface formation.

In line 52: 'Fujii and Kusunoki (1982) measured the sublimation and condensation at the ice sheet surface and suggested that the BFLs may be generated by the condensation of water vapour within the subsurface.'

In line 424: 'Based on previous crust studies and the research presented here, we believe that BFLs could form within the upper mm to cm of the snow surface.'

See line 424 in the revised manuscript.

Line 449-454: You don't have to hedge here – you have all the data in the ERA5 dataset you need to say something really concrete. Does a correlation between accumulation at Law Dome and 500 mb GPH produce fields similar to SOM1 or SOM2? If you think moisture transport is important, ERA5 has variables you can use in your analysis. You could also isolate times during accumulation hiatuses and

determine how atmospheric characteristics during hiatuses differ from baseline circulation patterns. Finally, if you want to relate BFLs to regional climate patterns, you should at least consider running correlations between BFLs in your ice core and different ERA5 variables. This will add a lot more specificity and detail to the discussion overall. Lots of this can be done in 10-15 minutes using online web apps. I use https://climatereanalyzer.org.

We are encouraged by the faith in our study! However, given our response to comments above about lines 309-310 in the submitted version of the manuscript, we think we still do have to hedge a bit here, and we think the next best step is to switch to the mechanistic investigation of formation using the SNOWPACK model.

In addition, an in depth study of accumulation at the Law Dome and Mount Brown South ice core sites during different synoptic types/situations is currently well-advanced and will be submitted elsewhere. This work will be very complementary to this study, and will certainly aid in the development of the BFL proxy, so we would prefer to develop this work in depth as a standalone manuscript, rather than add it as a small section here.

Line 465-469: This feels a little more like a thesis proposal than a paper. All you have to say is that if the mechanism behind BFL formation can be determined (perhaps through modeling) then they may have potential as a climate proxy.

We have reduced this section as suggested (deleted lines 465-469 in the original submitted version), however we would prefer to keep some of the discussion of the snowpack model, as we are actively working on using SNOWPACK to investigate BFL formation.

Conclusions: Since all of this repeats from previous material, I suggest removing, greatly reducing, or bolstering the discussion with additional information.

We agree. We removed the second and third paragraph in the Conclusion, which were repetitive about the importance and limitations of this study. For the remaining Conclusion parts, we also reduced the text about the results of this study to keep it concise and clear.

We also added new information in the last paragraph to point out the importance of this study:

'If the mechanism behind BFL formation can be determined (e.g. through modelling and analysis of longer records from different ice core sites) this may enable the development of a new proxy linking synoptic scale weather events to climate variability in ice core records.'

See Conclusion on page 24 in the revised manuscript.

Figure A2: This figure is very important – at least panel AH, which you conclude is most meaningful. I suggest putting this into the main body of the paper. This is potentially true for A3 and A4 as well. Also, not having the Udy et al. 2021 paper on hand, I wouldn't mind a plot showing the SOM1 and SOM2 patterns, at least in the appendix.

We have now moved Figure A2, A3 and A4 to the main body, as Figures 10, 11 and 12. Figure 10 and 11 are discussed in the paragraph which commences at line 324, as a part of the results for the analysis of the relationship between atmospheric processes and BFLs. Figure 12 is discussed in section 3.4, commencing at line 334 in the revised manuscript, as a part of the result for the analysis of the relationship between SOMs and BFLs.

See Figures 10, 11, 12 on page 19 and 20 in the revised manuscript.

We also added a schematic plot describing SOM1 and SOM2 patterns, which is discussed in line 339 and line 441 in the revised manuscript.

See figure 13, page 24 in the revised manuscript.

**Reviewer 2**

This paper takes a very detailed look at bubble free layers in the Law Dome ice cores and their potential drivers. A particular care is given to the identification of the timing of the occurrence of these layers, with respect to the season, which is essential for the correlation with climate variables like temperature, accumulation (or hiatus), and atmospheric circulation.

I am impressed with the level of details and care given in the paper. Not much more could have been done.

In the reading, I still struggle a bit with how things are presented, and in the section describing the exploration of mechanisms, some extra care could be put on describing hypotheses more clearly, and then, what data can be brought to support them or not.

A diagram/drawing showing the potential formation mechanisms would greatly help the reading of the paper.

We do appreciate that this would add to the reading of the paper but it is not something that we think we can add to this manuscript. As described in the introduction, this is a descriptive and exploratory study. The next stage of our work is to use the SNOWPACK model to investigate possible formation mechanisms using the analysis we present in this paper as a starting point. The data we present and analyse here is the occurrence of BFLs rather than formation. Currently, we don't know how frequently the mechanisms to produce BFLs occur without the preservation of a BFL (as preservation could be decoupled at times from formation, e.g. due to ablation of a newly formed layer). Equally, there could be a time lag between formation and apparent preservation. We clarified the text throughout the manuscript to ensure we differentiate clearly between BFL occurrence (studied in this work) and BFL formation/preservation (which we are currently commencing with the snowpack model, and intend to publish as a mechanistic study).

These are mostly cosmetic comments, on what is an excellent, thorough paper. I suggest minor revisions.

We thank the reviewer for their helpful and constructive comments.

line 35: bubble number density and paleoclimate: cite Fegyveresi 2016: https://agupubs.onlinelibrary.wiley.com/doi/full/10.1002/2015PA002851, also potentially:

Spencer, M. K., R. B. Alley, and J. J. Fitzpatrick (2006), Developing a bubble number-density paleoclimatic indicator for glacier ice, J. Glaciol., 52(178), 358– 364, doi:10.3189/172756506781828638.

Fegyveresi, J. M., R. B. Alley, M. K. Spencer, J. J. Fitzpatrick, E. J. Steig, J. W. C. White, J. R. McConnell, and K. C. Taylor (2011), Late-Holocene climate evolution at the WAIS Divide site, West Antarctica: Bubble number-density estimates, J. Glaciol., 57(204), 629– 638, doi:10.3189/002214311797409677.

Melt layers: consider citing Keegan 2014: www.the-cryosphere.net/8/1801/2014/ doi:10.5194/tc-8-1801-2014

We have added these references in the paper in the appropriate sections.

For Fegyveresi 2016, Spencer 2006 and Fegyveresi 2011.
See line 35-36  in the revised manuscript.

For Keegan 2014.
See line 45 in the revised manuscript.

line 97: missing space before citation.

We added a space there (before (Udy et al., 2021)).

See line 99 in the revised manuscript.

line 225: When you introduce the synoptic types, in section 2.5, you can include here the description of the two important types that you will use later, with a figure (potentially in the supplement if you feel you have too many figures) of the synoptic types, and their relationship with accumulation.

Please refer to the response to reviewer 1 above, which also  asks for a schematic or figure as well.

We have added a schematic plot describing SOM1 and SOM2 and showing the direction of circulation (see Figure 13). However, we put Figure 13 in the discussion section rather than the old line 225 (where we introduce the synoptic types).  There are 9 synoptic types (SOM1-SOM9) in total, only SOM1 and SOM2 are the two synoptic types related to BFLs. The other seven synoptic types (SOM3-SOM9) are not related to BFLs according to our study here. The detailed 9 synoptic types map are presented in Udy et al., 2021, and we mentioned and cited this publication (line100, line229) when introducing SOM. So we did not make a schematic plot for all 9 synoptic types. We added a schematic plot describing SOM1 and SOM2 in the

discussion to help readers understand the relationship between SOM1/SOM2 and BFLs.

Figure 13 is discussed in line 339 and line 441 in the revised manuscript.

See Figure 13, page 24 in the revised manuscript.

In general, I think that a description of the relationship between synoptic type and accumulation (or hiatus) is missing in a quantitative sense.
As I understand we could make a schematic:
 synoptic type --causes --> accumulation or hiatus --causes--> bubble free layer.  In this causal chain, you would find a causal relationship between synoptic type and BFL, but you would not be learning anything more about their formation than when you were looking at accumulation/hiatus.

We tried to detect the relationship between synoptic type and accumulation (or hiatus) quantitatively,  but the results were not as definitive as we had hoped. We think this is because we cannot (until we have done the mechanistic modelling currently underway with SNOWPACK) quantitatively define the formation date for BFLs. That is - we cannot with current data distinguish between the dates of formation and the dates of preservation (which may be different). With our current investigations using the SNOWPACK model, we can investigate the BFL formation mechanism, and then we will be able to relate the conditions needed for formation to larger scale phenomena, such as the synoptic types.

Figure 5: I can't see the grey for layer 2

Refer to response to reviewer 1 above.

We changed the grey bar into darker  grey. We also added an arrow to point out the position of the grey bar, which is narrow, as we wanted to demonstrate the diversity of BFL cases we see.

See Figure 6, page 12 in the revised manuscript.

Figure 7: the aws data is too tiny, Can you make it a bit thicker? parenthesis issue in caption

We made the AWS line a little thicker (not too much, as the thicker line is not able to show the tiny changes in snow surface height that are important to differentiate) and rectified the parenthesis issue.

See Figure 9, page 17 in the revised manuscript.

Yes, We have changed this into Table 4.

See line 337 in the revised manuscript.

worth adding that som1 is associated with high precip, or describing the significant modes a bit better

We have added a schematic plot describing SOM1 and SOM2 (Figure 13). We also described the relationship between SOM1, SOM2 and precipitation in line 442-448, as:

'The cyclones over the southern Indian and southwest Pacific Oceans are the main controlling factors of precipitation variation and extreme precipitation events for coastal East Antarctic regions like Law Dome (Uotila et al., 2011; Catto et al., 2015; Udy et al., 2022). During days with SOM1, the positive geopotential height anomaly blocks the passage of moist maritime air to Law Dome, leading to lower than average daily precipitation. In contrast when a SOM2 pattern predominates with the positive geopotential height anomaly located further east, moist maritime air masses from the mid-latitudes are channelled southward along the eastern boundary of the negative geopotential height anomaly to the East Antarctic coastline and plateau regions, which favours increased snowfall at Law Dome (Udy et al., 2022). '

Figure 13 is discussed in line 339 and line 441 in the revised manuscript.

See Figure 13, page 24 in the revised manuscript.

line 340+:trend: once you have analyzed the drivers, can you say something about the trends in the drivers themselves? Would we be expecting any trend?

This section is only describing an apparent (possibly spurious) trend in BFLs through time. We suggest this decreasing trend is an artificially significant trend, possibly due to the surface cores being more fragile and the BFLs in the surface core being more easily broken (and thus missed in the image analysis), rather than the formation of BFLs have a decreasing trend from 1990 to 2016. That is, it is possible that more fragile firn cores closer to the surface break preferentially at BFLs. We are currently analysing longer time series of BFL's from other ice core sites, and will be more comfortable discussing any trends in BFLs and their drivers once we have the longer

time series measured and analysed. As suggested, we can then revisit the drivers/mechanisms in future works to see whether they also have trends.

line 390: you have here a great comment about formation vs preservation of BFL.

I suggest that you restructure section 4 with the investigations of mechanisms, spelling them one by one, and detailing the supporting data/correlation, and the complicating factors: For instance, make a section on vapor flux upward condensing on the surface, as a formation mechanism, then show evidence for formation during snow-warmer-than-air times, during times of long surface exposure (your hiatus), seasonality, radiation, etc. And discuss evidence where it doesn't quite work.

Repeat this effort for other mechanisms of formation and preservation, aided by schematics, so that we can more clearly see where your thinking is.

We actually used this structure in a previous draft of this manuscript. After many rewrites and iterations, we decided that the structure suggested above was not the best way to structure the discussion. We think this is because we don't know the formation/preservation mechanism yet. We agree that the above suggestion is a logical way to think about the occurrence/formation/preservation of BFLs, however we found that it did not ultimately make for a linear and readable structure in the manuscript sense, largely because this is an exploratory study, and it is difficult to produce schematics of formation, when we have not yet explored formation in the model.

line 419 to 435: not sure it belongs in the paper. start of 4.4 to line 450: should go into the result section.

This section relates a lot to many of the comments from both reviewers requesting more quantitative analysis regarding BFL formation. As we have said, we think we need the model analysis first to do this. As this is the first paper of the first author's PhD project, we think it is worthwhile keeping this section, as it demonstrates clearly the work we are doing next. However, we have revisited this section to ensure relevance and brevity.

In the first section of 4.4, we removed the r and p values in the first sentence (as this is a repeat for the result section).

See line 436 in the revised manuscript.

Section 4.4: Give more info on the relationship between SOMs and accumulation/hiatus, temperature inversions, wind scours, etc..

We have given more information on the relationship between SOM1, SOM2 and precipitation/hiatuses, by adding a schematic plot describing SOM1 and SOM2 to show the direction of circulation (Figure 13). We also described the relationship between SOM1, SOM2 and precipitation in line 442-448, as:

'The cyclones over the southern Indian and southwest Pacific Oceans are the main controlling factors of precipitation variation and extreme precipitation events for coastal East Antarctic regions like Law Dome (Uotila et al., 2011; Catto et al., 2015; Udy et al., 2022). During days with SOM1, the positive geopotential height anomaly blocks the passage of moist maritime air to Law Dome, leading to lower than average daily precipitation. In contrast when a SOM2 pattern predominates with the positive geopotential height anomaly located further east, moist maritime air masses from the mid-latitudes are channelled southward along the eastern boundary of the negative geopotential height anomaly to the East Antarctic coastline and plateau regions, which favours increased snowfall at Law Dome (Udy et al., 2022). '

However the difference in spatial scales between the synoptic types and the smaller scale processes of wind scours and temperature inversions has not yet been explored (here or elsewhere). Again, we think this will be more appropriate in a subsequent work once we have the formation mechanism understood with modelling studies.

Figure 13 is discussed in line 339 and line 441 in the revised manuscript.

See Figure 13, page 24 in the revised manuscript.

Appendix A1: ERA is overestimating snowfall by a factor of 2. Did you consider scaling ERA, and plotting on figure A1 the scaled ERA to highlight the temporal accuracy of hiatuses, or perhaps even plot the derivative.

In A1,  ERA5 is the net accumulation (in metres water equivalent) and the AWS data is the measured snow surface height (including ablation, redistribution, settling etc). They are not really the same thing, so while scaling initially makes sense, it is ultimately misleading (we tried this previously, and felt it was more, rather than less confusing). However we adjusted the axes so that the two datasets can be more easily compared in a visual sense.

Figure A1 has been  moved into the body of the manuscript under the suggestion of Reviewer 1 and becomes Figure 4. Figure 4 is discussed at line 184 in the revised manuscript.

See Figure 4, page 9.